# Avoiding Structural Pitfalls: Self-Supervised Low-Rank Feature Tuning for Graph Test-Time Adaptation

**Haoxiang Zhang**[*]      *haz140@ucsd.edu*
*Halıcıoğlu Data Science Institute*
*University of California, San Diego*

**Zhuofeng Li**[*]      *zhuofengli12345@gmail.com*
*Independent*

**Qiannan Zhang**      *qiz4005@med.cornell.edu*
*Weill Cornell Medicine College*
*Cornell University*

**Ziyi Kou**      *zkou@nd.edu*
*Department of Computer Science and Engineering*
*University of Notre Dame*

**Juncheng Li**      *jcli@cs.ecnu.edu.cn*
*School of Computer Science and Technology*
*East China Normal University*

**Shichao Pei**[†]      *shichao.pei@umb.edu*
*Department of Computer Science*
*University of Massachusetts Boston*

**Reviewed on OpenReview:** *https://openreview.net/forum?id=yiS6q42LLt*

## Abstract

Pre-trained graph neural networks (GNNs) have demonstrated significant success in leveraging large-scale graph data to learn transferable representations. However, their performance often degrades under distribution shifts, particularly in real-world scenarios where labels of test data are unavailable. To address this challenge, we propose **G**raph **O**ptimization via **A**ugmented **T**ransformations (GOAT), a novel self-supervised test-time tuning paradigm that adapts pre-trained GNNs to distribution-shifted test data by focusing exclusively on node feature transformations. By avoiding complex and often suboptimal graph structure transformations, GOAT overcomes the limitations of existing data-centric methods. To further address the issue of transformation collapse, where feature transformations converge to trivial solutions — such as when the test-time learned data-centric transformation degenerates into a constant or identity mapping across different inputs, we introduce a parameter-efficient low-rank adapter that generates diverse transformations tailored to individual input graphs. This design not only enhances adaptation performance but also improves interpretability by avoiding modifications to the graph structure. Through extensive experiments on six real-world datasets with diverse distribution shifts, we demonstrate that GOAT achieves consistent performance improvements across different pre-trained GNN backbones, outperforming state-of-the-art test-time adaptation methods.

---

[*]Equal contribution.
[†]Corresponding author.

| Dataset | OGB-ArXiv | | | Elliptic | | |
|---|---|---|---|---|---|---|
| **Description** | Open-world dataset of academic papers, the graph evolves as new papers are cited. | | | A dataset of transactions labeled as licit or illicit, influenced by market conditions. | | |
| **Split** | Year Slice | Accuracy | Degrade | Time Slice | F1 score | Degrade |
| **Train** | before 2011 - 2011 | 47.88% | | $7^{th}$ - $11^{th}$ | 90.12% | |
| **Val** | 2011 - 2014 | 44.46% | -9.96% | $12^{th}$ - $17^{th}$ | 78.75% | -39.17% |
| **Test** | 2014 - 2020 | 38.92% | | $17^{th}$ - $49^{th}$ | 50.95% | |

Table 1: The showcase indicates a significant decrease in the node classification performance of the pre-trained GCN on the OGB-ArXiv (Hu et al., 2020) and Elliptic (Pareja et al., 2020) datasets in an OOD setting where graph data is generated from different time intervals. For OGB-ArXiv, the year ranges from before 2011 to 2020; for Elliptic, from the $7^{th}$ snapshot (when the dark market crackdown occurred) to the $49^{th}$ snapshot. Performance degrades noticeably during validation and testing as time progresses.

# 1 Introduction

Graph pre-training has become a powerful technique for capturing and preserving information from large-scale upstream graph data (Kipf & Welling, 2022; Hamilton et al., 2017; Veličković et al., 2018), enabling graph neural networks (GNNs) to learn rich representations that can be effectively transferred to a variety of downstream graph tasks. However, the performance of pre-trained GNNs is often hindered by distribution shifts (Yehudai et al., 2021; Li et al., 2022a; Song & Wang, 2022; Zhu et al., 2021), particularly in real-world scenarios where the labels of the test data are unavailable. This presents a significant obstacle to the practical deployment of GNNs, as their performance can degrade substantially under such conditions. For example, Table 1 illustrates that the performance of a pre-trained GNN deteriorates as the distribution of the test data changes over time.

Adapting pre-trained GNNs to such unlabeled and distribution-shifted test scenarios is therefore a crucial yet under-addressed problem. Several methods have been proposed to deal with distribution shift at *training time*, including invariant risk minimization (Arjovsky et al., 2019; Wu et al., 2023), domain-invariant learning (Muandet et al., 2013; Li et al., 2022b), and invariant representation learning (Wu et al., 2022; Chen et al., 2022b). These approaches attempt to learn features that remain stable across environments, assuming access to labeled data in the target domain during training. Unfortunately, in many practical settings, test-time data is both unlabeled and unavailable at training time, rendering these methods inapplicable for adapting pre-trained models at test time.

Test-time adaptation (TTA) offers a promising alternative. Model-centric TTA methods, such as updating classifier heads (Wang et al., 2022) or fine-tuning the full model (Zhang et al., 2024; Wang et al., 2021), aim to leverage the generalization capabilities of pre-trained models. However, these approaches often struggle to adapt effectively to unseen or shifted distributions, as they heavily depend on parameters and statistics learned from the training data (Hendrycks & Dietterich, 2019; Arjovsky et al., 2019). Moreover, fine-tuning the entire model can be computationally expensive, making it impractical in many real-world resource-constrained scenarios. In contrast to model-centric approaches, data-centric TTA methods (Jin et al., 2023; Chen et al., 2022a; Zhang et al., 2024) shift the focus from updating model parameters to refining the test graph itself. By transforming node features and graph structure, these methods aim to better align the test data with the representation space of the pre-trained model. This strategy has shown considerable promise in improving model performance under distribution shifts.

Despite their effectiveness, data-centric methods typically rely on simultaneous transformations of both node features and graph structures, as adapting node features alone often leads to suboptimal performance. However, this requirement introduces two key challenges. First, identifying and optimizing appropriate graph structure transformations is inherently difficult due to the vast and continuous search space. Inaccurate structure modifications can result in poorly refined test graphs, ultimately degrading adaptation performance. Second, altering the graph structure reduces the interpretability of the adaptation process by obscuring the relationship between the original and transformed graphs, making it difficult to trace how specific structural changes influence model predictions.

While one might consider avoiding structure transformations by focusing exclusively on node features, this approach introduces its own difficulties. In particular, it risks both reduced performance and transformation collapse, where the transformation function becomes ineffective, often converging to a constant transformation or identity mapping that fails to facilitate meaningful adaptation.

**Present work.** To address these limitations and overcome the associated challenges, we propose a novel self-supervised test-time tuning paradigm - **G**raph **O**ptimization via **A**ugmented **T**ransformations (GOAT) that enables the pre-trained GNN to dynamically adapt to unseen test distributions without requiring access to test labels, source training data, or training details. 1) To avoid modifying the test graph structure, our method GOAT focuses exclusively on node feature transformations. However, naïvely transforming node features is insufficient for effective adaptation. To overcome this, we introduce a self-supervised strategy that estimates optimal node feature transformations by leveraging a set of augmented test graphs. By measuring the embedding discrepancies between augmented graphs before and after transformation, our method learns a generalizable transformation function tailored to the test distribution. 2) To address the transformation collapse issue, we propose a low-rank adapter that generates unique transformations for different input graphs while effectively utilizing the knowledge encoded in the pre-trained GNN. This approach ensures that, during each learning iteration, the augmented graphs undergo diverse transformations by the dynamic adapter, preventing uniform transformations across all graphs and preserving the diversity of the augmented data. With the learned low-rank adapter, the model's interpretability becomes more intuitive compared to interpreting transformations of the graph structure. To summarize, our main contributions are as follows:

- Our work centers on data-centric test-time adaptation and introduces a novel self-supervised test-time tuning paradigm, GOAT, that leverages node features exclusively, bypassing the complexity of optimizing for structure transformations.

- We propose a parameter-efficient low-rank adapter to address the transformation collapse issue while enhancing model interpretability.

- Extensive experiments on real-world datasets with diverse distribution shifts demonstrate consistent performance improvements across various backbones.

## 2 Related Work

**Distribution Shift on Graphs.** Graph-structured data often exhibits distribution-shift phenomena (Song & Wang, 2022; Li et al., 2022a). To tackle this challenge, researchers have proposed methods for learning invariant representations (Wu et al., 2023; Arjovsky et al., 2019; Wu et al., 2022; Chen et al., 2022b; Li et al., 2022b; Muandet et al., 2013), generalizing pre-trained GNNs (Zhu et al., 2021; Li et al., 2022a; Song & Wang, 2022; Hu et al.; Zhao et al., 2021), detecting OOD instances (Zellinger et al., 2022; Guo et al., 2023; Huang et al., 2024). Most of these approaches often require access to multiple source domains, rely on specific model architectures and train-time paradigms, or may lead to performance degradation. For a thorough review, we refer the readers to two recent surveys (Wu et al., 2024; Liu & Ding, 2024).

**Graph Test-time Adaptation.** Graph test-time adaptation (GTTA) aims to adapt pre-trained models to the test distribution without requiring labeled data or modifying the model's parameters (Chen et al., 2022a). Existing proposed methods can be broadly categorized into the following two classes: data-centric and model-centric. *(1) Model-centric.* These approaches center on the learning process or the design of the graph model. SLAPS (Fatemi et al., 2021), TTT (Sun et al., 2020), TTT++ (Liu et al., 2021), and Tent (Wang et al., 2021), update the whole pre-train model or some specific layers. In addition, GraphTTA (Chen et al., 2022a) and GT3 (Wang et al., 2022) tune a new classifier head expecting better prediction. However, these approaches have limitations such as reliance on specific architectures, over-smoothing, or being preoccupied with how to select negative samples for contrastive learning. These issues direct our attention to data-centric approaches. *(2) Data-centric.* These recently emerging approaches emphasize the manipulation of input graphs. GTRANS (Jin et al., 2023) both modify the adjacency matrix and node feature to empower graph representation learning. GraphCTA (Zhang et al., 2024) uses a memory bank to keep the best neighbor

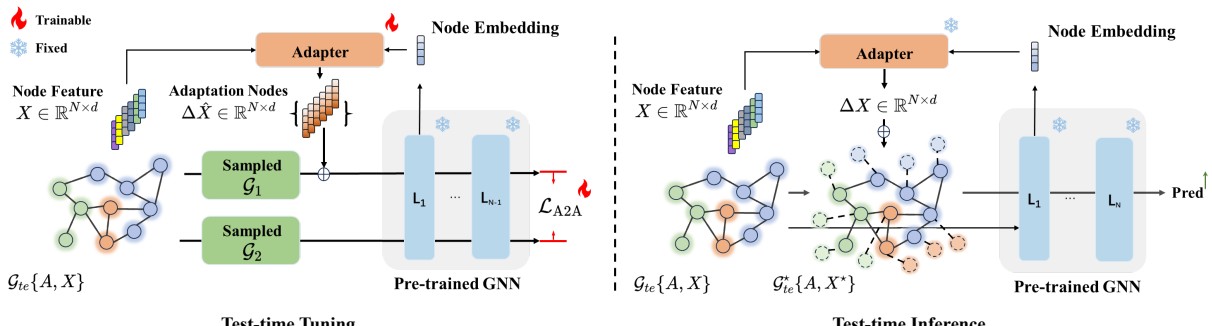

Figure 1: Overview of the proposed method GOAT under two augmented views. In the **test-time tuning**, multiple augmented views of a single test graph are generated by extracting induced subgraphs. These views are passed through the fixed pre-trained GNN, with their input node features modified by an adaptation term $\Delta\hat{X}$. A self-supervised loss $\mathcal{L}_{A2A}$ is used to train the adapter, encouraging alignment between the embeddings of one view with node feature transformation and another without. During the **test-time inference** phase, all parameters are frozen. The node feature and its embedding after certain layers of the pre-trained GNN will be fed to the adapter. The best adaptation $\Delta X$ generated from the adapter is added to the input features of the test graph and passed through the pre-trained GNN, resulting in improved prediction performance.

structure. Whereas, GTRANS requires the transformation of graph structure, leading to the suboptimal structure refinement, while GraphCTA still needs to update the whole pre-trained model's parameters. Graph prompt tuning (Fang et al., 2024) inspires a succinct direction; however, it emphasizes the task discrepancy rather than the distribution shift and relies on the availability of label sets for downstream tasks. These challenges guide us toward a more universal test-time tuning paradigm.

## 3 Methodology

In this section, we delve into our proposed paradigm GOAT and Figure 1 provides the framework overview.

### 3.1 Self-supervised Graph Test-time Adaptation from Augmentations

A graph is represented by $\mathcal{G}$ with a set of nodes $\mathcal{V}$ and a set of edges $\mathcal{E}$. For any input graph data $\mathcal{G} = (A, \mathcal{X})$, the adjacency matrix derived from a graph is denoted by $A \in \mathbb{R}^{N \times N}$. If $v_i, v_j \in \mathcal{V}$ and $(v_i, v_j) \in \mathcal{E}$, then $A_{ij}$ is one, otherwise it is zero and $\mathcal{X} = \{x_v \mid v \in \mathcal{V}\}$ is the node features. Apart from these, each node in the graph has a label $y_v$, and the label of the whole graph can be represented as a vector $\mathcal{Y}$. In a local view, we define an induced graph $\mathcal{G}_S = (A_S, \mathcal{X}_S)$, where $S \subseteq \mathcal{V}$, $A_S = [A_{vu} \mid v, u \in S]$, and $\mathcal{X}_S = \{x_v \mid v \in S\}$. Its corresponding label is $\mathcal{Y}_S$.

**Graph Pre-train.** During train time, a GNN $f_\theta(\mathcal{G})$ parameterized by $\theta$ is optimized by the input-target pairs $(\mathcal{G}_S, \mathcal{Y}_S)$ sampled from training data $\widetilde{D}_{tr} = \{\mathcal{G}_{tr}, \mathcal{Y}_{tr}\}$, which minimizes the empirical risk:

$$\theta^\star = \arg\min_\theta \mathbb{E}_{(\mathcal{G}_S, \mathcal{Y}_S)} \mathcal{L}_{tr}(f_\theta(\mathcal{G}_S), \mathcal{Y}_S). \tag{1}$$

**Distribution Shift.** It's a key challenge in real-world applications that the training $\widetilde{D}_{tr}$ and test graph $\mathcal{G}_{te}$ together with its label $\mathcal{Y}_{te}$ are drawn from different probability distributions. Formally, if we denote the source distribution as $P_{tr}(\mathcal{G}, \mathcal{Y} | \widetilde{D}_{tr})$ and the test distribution as $P_{te}(\mathcal{Y}_{te} | \mathcal{G}_{te})$, distribution shift is defined via a divergence measure $d(\cdot, \cdot)$ as:

$$\Delta = d\big(P_{tr}(\mathcal{G}, \mathcal{Y} | \widetilde{D}_{tr}), \ P_{te}(\mathcal{Y}_{te} | \mathcal{G}_{te})\big) > \epsilon$$

Here, $\epsilon$ is assumed to be a relatively large threshold, meaning that the distributional gap is substantial enough to hinder direct generalization of the source-trained model. This mismatch means a model optimized

on the source distribution may not generalize effectively. The problem is particularly acute when test labels are unavailable, which makes determining the true test-time distribution more difficult and is a core challenge we aim to address.

**Graph Test-time Adaptation.** At test time, for arbitrary graph $\mathcal{G}_{te}$, when the label $\mathcal{Y}_{te}$ is available, following the data-centric paradigm (Jin et al., 2023; Zhang et al., 2024), an extra transformation of the input graph data should satisfy the following objective:

$$\psi^{\star} = \arg\min_{\psi} \mathbb{E}_{\mathcal{V} \atop (\mathcal{G}_{te}, \mathcal{Y}_{te})} \mathcal{L}_{te}(f_{\theta^{\star}}(g_{\psi}(\mathcal{G}_{te})), \mathcal{Y}_{te}), \tag{2}$$

denoted that for parameter efficiency, the parameters $\theta^{\star}$ of the pre-trained GNN $f(\mathcal{G})$ is fixed during tuning the graph transformation $g(\mathcal{G}) : \mathbb{G} \to \mathbb{G}$ parameterized by $\psi$ and the number of parameters $|\psi|$ should be significantly smaller than $|\theta^{\star}|$. During the prediction stage, $\mathcal{L}_{te}$ represents the likelihood loss, while in the representation learning stage, $\mathcal{L}_{te}$ can be defined as the $L_2$ norm. Since the pre-trained $\theta^{\star}$ are fixed at test-time and the test graph is the focus of this paper, we drop the subscript in $\mathcal{G}_{te}$ and $f_{\theta^{\star}}$ to simplify notations in the rest of the paper.

When test labels $\mathcal{Y}_{te}$ are unavailable, we reframe the original task of predicting $\mathcal{Y}_{te}$ as estimating the optimal representation $z^{\star} = f(g_{\psi}^{\star}(\mathcal{G}_{te}))$, where $f$ is the fixed pre-trained GNN encoder, and $g_{\psi}$ is a transformation function applied at test time. To estimate the optimal representation $z^{\star}$, we use *proxies*, a strategy akin to estimating an unknown quantity (e.g., room temperature) by averaging multiple noisy measurements. According to the law of large numbers, this can yield a reasonably accurate approximation of the ground truth. Since we do not modify the pre-trained model or access ground-truth labels, we optimize $g_{\psi}$ using *proxy representations* $\hat{z}$ obtained from augmented versions of the test graph, denoted $\hat{\mathcal{G}}_{te}$:

$$\hat{z} = f(\hat{\mathcal{G}}_{te}). \tag{3}$$

Using these proxies, we formulate the objective to learn $g_{\psi}$ as:

$$\psi^{\star} = \arg\min_{\psi} \mathbb{E}_{(\mathcal{G}_{te}, \hat{z})} \left\| f(g_{\psi}(\mathcal{G}_{te})) - \hat{z} \right\|^2, \tag{4}$$

where the $L_2$ norm is used as the loss $\mathcal{L}_{te}$ in the representation learning stage. Since we use $L_2$ loss, it suffices that the proxy $\hat{z}$ satisfies $\mathbb{E}_{\hat{z}}[\hat{z}] = \mathbb{E}_z[z^{\star}]$.

To ensure generalization, $g_{\psi}$ must be capable of transforming both the original test graph $\mathcal{G}_{te}$ and its various augmented versions $\hat{\mathcal{G}}_{te}$. To this end, we train $g_{\psi}$ on $\mathcal{G}_{te}$ and generate proxies from $\hat{\mathcal{G}}_{te}$. Importantly, to prevent $g_{\psi}$ from collapsing to the identity function, the input graph passed to $g_{\psi}$ must be different from the one that generates $\hat{z}$. Specifically, let $\widetilde{\mathcal{D}}_{aug} = \{\mathcal{G}_1, \mathcal{G}_2, \ldots, \mathcal{G}_{|\tau|}\}$ be a set of $|\tau|$ augmented graphs derived from $\mathcal{G}_{te}$. We train $g_{\psi}$ to transform each $\mathcal{G}_p \in \widetilde{\mathcal{D}}_{aug}$, while selecting different graphs $\mathcal{G}_q \in \widetilde{\mathcal{D}}_{aug} \setminus \mathcal{G}_p$ as the proxies. The optimization objective becomes (proofs can be found in Appendix A):

$$\mathcal{L}_s(\psi) = \mathbb{E}_{\widetilde{\mathcal{D}}_{aug}} \left[ \frac{1}{\binom{|\tau|}{2}} \sum_{p=1}^{|\tau|} \sum_{\substack{q=1 \\ q \neq p}}^{|\tau|} \left\| f(g_{\psi}(\mathcal{G}_p)) - f(\mathcal{G}_q) \right\|^2 \right], \tag{5}$$

subject to the constraint:

$$\mathbb{E}_{\nu} \left[ f(g_{\psi}(\mathcal{G}_i)) - f(\mathcal{G}_i) \right] = 0, \quad \forall \mathcal{G}_i \in \widetilde{\mathcal{D}}_{aug}. \tag{6}$$

To efficiently construct $\widetilde{\mathcal{D}}_{aug}$, We apply a subgraph sampling strategy, similar in principle to methods (Rong et al., 2020; Hamilton et al., 2017), to generate sufficient sub-views of augmentation of the test graph. Note that the graph transformation $g_{\psi}$ here is applied exclusively to the node features of these subgraphs. The loss $\mathcal{L}_s(\mathcal{G}_p, \mathcal{G}_q)$ is symmetric with respect to its inputs, swapping $\mathcal{G}_p$ and $\mathcal{G}_q$ leaves the loss unchanged. This symmetry ensures that the adaptation process is invariant to the input order, thereby enhancing the robustness of the data-centric optimization approach.

### 3.2 Node Feature Transformation with Low-rank Cross-attention Adapter

During optimizing the transformation $g_\psi$ with $\mathcal{L}_s$ in Eq.(5), collapse mapping occurs when each augmented graph receives the same transformation solution, leading to a failure in producing meaningful input-specific adaptations. We define this phenomenon as *transformation collapse*, a failure mode where the adapter converges to a trivial function that fails to generate distinct outputs for different inputs. This includes two primary forms: a) Constant Collapse: Where the data-centric transformation $g_\psi$ maps all different inputs to the same output ($g_\psi(\mathcal{G}_p) = \mathcal{G}', \forall \mathcal{G}_p$). b) Identity Collapse: Where $g_\psi$ simply returns the input ($g_\psi(\mathcal{G}_p) = \mathcal{G}_p$). To address this issue, we design a low-rank cross-attention adapter to ensure that distinct inputs $\mathcal{G}$ produce unique outputs $g_\psi(\mathcal{G})$, thereby preventing all augmented graphs from being mapped to the same point in the pre-trained GNN's representation space from a data-centric perspective.

For an arbitrary graph $\mathcal{G} = (A, \mathcal{X})$, where $A \in \mathbb{R}^{N \times N}$ represents the adjacency matrix encoding the edge relationships among $N$ nodes, and $\mathcal{X} \in \mathbb{R}^{N \times d}$ denotes the node feature matrix with $d$-dimensional features for each node, data-centric graph adaptation assumes the existence of a transformed graph $\mathcal{G}^\star = g_{\psi^\star}(\mathcal{G})$, which ensures that the graph achieves the best possible performance when applied to the pre-trained GNN $f(\cdot)$, by bridging the gap between the test graph and the knowledge learned by $f(\cdot)$. Specifically, a graph transformation that only transforms node features is defined as:

$$g(\mathcal{G}) = (A, \mathcal{X} \oplus \Delta X), \tag{7}$$

where $\Delta X \in \mathbb{R}^{N \times d}$ denotes continuous learnable parameters with the same shape as the node features. Here, $\oplus$ denotes element-wise addition.

However, simply adding $\Delta X$ cannot solve the collapse mapping issue. To better solve this issue and integrate global and local information of test graphs into $\Delta X$ while fully leveraging the knowledge learned by the pre-trained GNN, we design the following adapter incorporating the pre-trained GNN to generate $\Delta X$ dynamically.

In GNNs, learning of node representations typically relies on the aggregation of neighboring nodes. Given a graph $\mathcal{G} = (A, \mathcal{X})$, an $L$-layer GNN can be represented as:

$$H^{(l)} = \sigma(AGG^{(l)}(A, H^{(l-1)})), \quad l = 1, 2, \ldots, L, \tag{8}$$

where $H^{(l)} \in \mathbb{R}^{N \times d^{(l)}}$ represents the node representation at the $l$-th layer in pre-trained GNN with $H^{(0)} = \mathcal{X}$, $AGG^{(l)}(\cdot)$ is the aggregation function at the $l$-th layer, and $\sigma$ is the activation function. In our experiments, the ReLU activation function is employed: $\sigma(x) = x^+$.

Although GNNs effectively capture local structural information through neighborhood aggregation, they face two key limitations. First, they primarily focus on local interactions and lack mechanisms to model global structural context. Second, modeling long-range dependencies requires stacking multiple layers, which significantly increases computational cost and often leads to over-smoothing. To address these challenges, we propose the Low-Rank Adapter (LRA), a lightweight module designed to incorporate global context efficiently via attention over a compact set of virtual nodes.

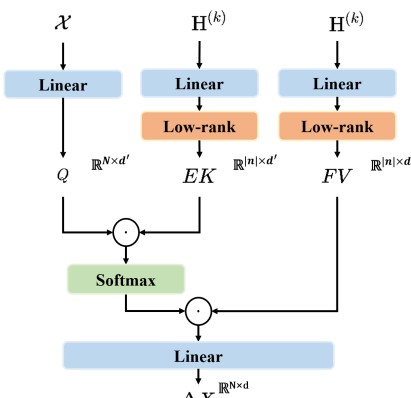

Figure 2: Structure overview of Low-Rank Adapter.

Specifically, the LRA operates on the $k$-th output of the first $k$ layers of the GNN with the original node features to obtain an $k$-hop neighbor-aware node representation $H^{(k)} \in \mathbb{R}^{N \times d^{(k)}}$. Then we introduce Low-Rank projection : $\mathbb{R}^{N \times d'} \to \mathbb{R}^{|n| \times d'}$ (usually $|n| \ll N$), which is a low-rank projection function that projects nodes into a latent space represented by few virtual nodes, striving for maximal rank density and minimal computational overhead, where $d'$ is the attention dimension in LRA. Let $E, F \in \mathbb{R}^{|n| \times N}$ denote the key and value's low-rank transition matrices of the virtual nodes. The

core of LRA is a self-attention mechanism, which enables each node to attend to a globally-aware summary of the graph via virtual nodes. The computation process of LRA is:

$$\Delta X = (\text{Softmax}(\frac{Q(EK)^T}{\sqrt{d'}}) \cdot (FV))W_X, \tag{9}$$

where $W_X \in \mathbb{R}^{d' \times d}$ is a learnable output projection matrix, and $Q, K, V$ are obtained via:

$$Q = \mathcal{X}W_Q, \quad K = H^{(k)}W_K, \quad V = H^{(k)}W_V. \tag{10}$$

Here, $W_Q \in \mathbb{R}^{d \times d'}$, $W_K \in \mathbb{R}^{d^{(k)} \times d'}$, and $W_V \in \mathbb{R}^{d^{(k)} \times d'}$ are learnable weight matrices. By introducing the low-rank transition matrices $E$ and $F$, LRA can compute long-range dependencies between nodes with linear time complexity and generate $\Delta X \in \mathbb{R}^{N \times d}$ that contain local-global contextual information. The time complexity of the LRA, assuming $k = 1$, is $O(Nd')$.

With LRA, both interpretability and robustness are improved. The output $\Delta X$ can be directly analyzed, making it more interpretable than structural graph transformations. Additionally, in the test-time adaptation setting, LRA helps prevent transformation collapse: when applied to different graphs $g(\mathcal{G}_q)$ and $g(\mathcal{G}_p)$, their outputs are less likely to degenerate to the same representation, thus maintaining representational diversity.

### 3.3 Optimization

The constraint in Eq.(5) requires the same input graph for both $f(g_\psi(\cdot))$ and $f(\cdot)$, which can easily result in the mapping collapse issue. As transformation $g(\mathcal{G})$ can be formulated as $(A, \mathcal{X} + \Delta X)$, the constraint is equivalent to:

$$\arg\min_{\Delta X} \mathbb{E}_\mathcal{V} \| f(A, \mathcal{X} + \Delta X) - f(A, \mathcal{X}) - f(A, \Delta X) \|^2$$
$$\text{s.t.} \quad \mathbb{E}_\mathcal{V}[f(A, \Delta X)] = 0, \tag{11}$$

where $f(A, \Delta X)$ serves as a regularization. We define $g_\psi(f(\mathcal{G})) = f(A, \mathcal{X}) + f(A, \Delta X)$ as the transformation on node embeddings. In this form, it can be seen as encouraging $g_\psi$ to learn an isomorphic mapping with $f$, ensuring that the adapter's transformations are consistent with those of the GNN, i.e., $g_\psi \circ f \approx f \circ g_\psi$. This consistency facilitates the rapid optimization and performance consistency of the transformation $g_\psi$ across different designs of $f$. With further relaxing the constraint in Eq.(11), we derive the following consistency loss and a regularization loss. Optimizing the two loss functions jointly can achieve the same effect as optimizing the constraints in Eq.(6) with the same $|\tau|$ augmented graphs in Eq.(5):

$$\mathcal{L}_c(\psi) = \mathbb{E}_{\mathcal{V} \atop 1 \le p \le |\tau|} [\frac{1}{|\tau|} \sum_{|\tau|} \| f(g_\psi(\mathcal{G}_p)) - g_\psi(f(\mathcal{G}_p)) \|^2]$$
$$\mathcal{L}_R(\psi) = \mathbb{E}_{\mathcal{V} \atop 1 \le p \le |\tau|} [\frac{1}{|\tau|} \sum_{|\tau|} \| f(A_p, \Delta X_p) \|^2], \tag{12}$$

where $A_p$ is the adjacent matrix of $\mathcal{G}_p \in \widetilde{D}_{aug}$, $\Delta X_p$ is generated from LRA with the input of $\mathcal{G}_p$.

The overall loss function used for training is defined as follows:

$$\mathcal{L}_{A2A} = \alpha\lambda(\mathcal{L}_s + \mathcal{L}_c) + (1 - \alpha)\mathcal{L}_R, \tag{13}$$

where $\alpha$ and $\lambda$ are hyperparameters that control the importance of each objective. $\alpha$ is a hyperparameter in the range $(0, 1)$; $\lambda$ is a positive hyperparameter.

## 4 Experiments

### 4.1 Adaptation on Data with Distribution Shift

**Datasets.** We evaluate GOAT 's performance on three types of distribution shifts across six benchmark datasets, following the experimental settings of EERM (Wu et al., 2023). The dataset statistics, along with

Table 2: Average classification performance (%) on the test graphs. The best performance on each dataset with a specific backbone is indicated in bold, the second-best method is underlined, and "Avg." indicates the average ranking of the same method across all six datasets under the same backbone. OOM indicates an out-of-memory error on a 24 GB GPU. $^{\uparrow}$/* denotes that GOAT significantly outperforms GTrans and ERM using the Wilcoxon signed-rank test at the significance level of $p < 0.05$, respectively, across all datasets.

| Backbone | Method | Artificial Transformation | | Temporal Evolution | | Cross-Domain | | Avg. |
| | | Amz-Photo | Cora | Elliptic | OGB-Arxiv | Twitch-E | FB-100 | |
|---|---|---|---|---|---|---|---|---|
| GCN | ERM | 92.78±1.34 | 93.92±0.64 | 54.13±1.18 | 36.89±0.67 | 56.84±1.13 | 53.95±0.77 | 4.7 |
| | EERM | 94.24±0.40 | 87.36±0.86 | 53.15±0.01 | OOM | 57.25±0.42 | 54.03±0.80 | 5.2 |
| | Tent | 93.84±1.53 | 91.64±2.37 | 46.72±0.06 | 39.34±2.76 | 60.01±0.95 | 54.11±1.50 | 4.2 |
| | GCTA | 91.43±1.74 | 93.13±2.02 | 55.82±3.50 | 37.27±3.46 | 60.10±0.95 | 54.11±1.49 | 3.7 |
| | GTRANS | 94.32±1.34 | 94.76±1.94 | 55.07±3.61 | 40.45±1.76 | 60.37±1.44 | 54.17±1.23 | 1.8 |
| | GOAT | **94.35±1.32** | **94.79±1.36** | **55.83±3.81** | *39.44±2.02 | *60.15±1.30 | **54.19±2.04** | **1.3** |
| SAGE | ERM | 87.79±1.74 | 99.62±0.09 | 50.11±0.39 | 37.52±0.66 | 59.20±0.14 | 54.09±0.40 | 5.2 |
| | EERM | 95.76±0.11 | 99.76±0.21 | 60.43±0.29 | OOM | 60.09±0.25 | OOM | 5.2 |
| | Tent | 95.23±1.52 | 99.71±0.17 | 50.25±3.28 | 39.56±1.49 | **62.05±0.22** | 55.11±0.55 | 3.0 |
| | GCTA | 96.86±1.11 | 99.85±0.06 | 66.92±2.33 | 33.67±3.25 | 62.05±0.24 | 55.11±0.56 | 3.2 |
| | GTRANS | 97.09±1.13 | 99.81±0.16 | 63.04±6.39 | 39.74±1.14 | 61.97±0.34 | 55.07±0.59 | 2.5 |
| | GOAT | *92.54±2.51 | *99.89±0.10 | *67.92±5.56 | *39.52±1.03 | *61.91±0.28 | *55.61±0.30 | 2.5 |
| GAT | ERM | 94.92±2.33 | 95.99±0.88 | 49.49±1.51 | 37.92±0.68 | 57.36±0.30 | 48.25±1.55 | 3.8 |
| | EERM | 94.07±1.32 | 79.35±8.90 | 54.27±2.42 | OOM | 56.27±0.37 | 52.46±2.02 | 3.7 |
| | Tent | 94.96±0.87 | 93.54±3.50 | 55.29±5.22 | 37.41±5.20 | 58.93±1.50 | 51.22±1.99 | 5.3 |
| | GCTA | 94.72±1.73 | **96.03±1.76** | 56.00±10.11 | 37.8 6±2.17 | 58.83±1.59 | 51.22±1.98 | 3.2 |
| | GTRANS | **95.14±0.70** | 95.46±1.96 | **62.56±4.22** | 37.52±2.68 | 58.84±1.49 | 51.27±1.91 | 2.5 |
| | GOAT | 94.69±0.63 | 94.72±2.83 | *60.33±4.83 | $^{\uparrow}$*41.13±1.96 | 58.95±1.50 | $^{\uparrow}$*54.20±1.10 | 2.3 |
| GPR | ERM | 84.81±3.71 | 83.98±1.72 | 48.96±1.05 | 40.91±0.28 | 57.25±0.66 | 54.36±0.27 | 4.1 |
| | EERM | 90.87±0.52 | 87.16±2.39 | 60.08±0.03 | OOM | 58.75±0.29 | 54.21±0.42 | 4.0 |
| | Tent* | - | - | - | - | - | - | - |
| | GCTA | 91.96±0.75 | 92.75±2.48 | 66.36±3.67 | 44.44±0.70 | 59.97±0.62 | 54.63±0.77 | 2.6 |
| | GTRANS | 91.97±0.84 | 92.70±2.46 | 68.54±5.56 | 45.64±0.61 | 59.84±0.89 | 54.48±0.66 | 2.4 |
| | GOAT | *91.98±0.83 | *92.79±2.74 | *66.47±6.44 | *44.78±0.69 | *60.00±0.65 | $^{\uparrow}$*55.23±0.43 | 1.7 |

* Tent cannot be applied to models that do not contain batch normalization layers.

Table 3: Summary of the experimental datasets that entail diverse distribution shifts.

| Dataset | Distribution Shift | #Nodes | #Edges | #Classes | Train/Val/Test Split | Metric | Adapted From |
|---|---|---|---|---|---|---|---|
| Cora | Artificial Transformation | 2,703 | 5,278 | 10 | Domain-Level | Accuracy | (Yang et al., 2016) |
| Amazon-Photo | | 7,650 | 119,081 | 10 | Domain-Level | Accuracy | (Shchur et al., 2018) |
| Elliptic | Temporal Evolution | 203,769 | 234,355 | 2 | Time-Aware | F1 Score | (Pareja et al., 2020) |
| OGB-ArXiv | | 169,343 | 1,166,243 | 40 | Time-Aware | Accuracy | (Hu et al., 2020) |
| Twitch-Explicit | Cross-Domain Transfer | 1,912 - 9,498 | 31,299 - 153,138 | 2 | Domain-Level | ROC-AUC | (Rozemberczki et al., 2021) |
| Facebook-100 | | 769 - 41,536 | 16,656 - 1,590,655 | 2 | Domain-Level | Accuracy | (Traud et al., 2012) |

a breakdown of these three distinct types of distribution shifts, are presented in Table 3: (1) **Artificial Transformation** for Cora (Yang et al., 2016) and Amazon-Photo (Shchur et al., 2018), where node features are replaced by synthetic features. (2) **Cross-Domain** transfers for Twitch-E (Rozemberczki et al., 2021) and FB-100 (Traud et al., 2012), involving graphs from different domains. (3) **Temporal Evolution** for Elliptic (Pareja et al., 2020) and OGB-ArXiv (Hu et al., 2020), utilizing dynamic datasets with natural evolving characteristics. The datasets are split into training/validation/test sets with ratios of: 1/1/8 for Cora and Amazon-Photo; 1/1/5 for Twitch-E; 3/2/3 for FB-100; 5/5/33 for Elliptic; and 1/1/3 for OGB-ArXiv.

**Baselines.** GOAT is compared with four baselines: empirical risk minimization ERM, test-time training method Tent (Wang et al., 2021), memory-bank-based method GraphCTA(GCTA) (Zhang et al., 2024), the train-time state-of-the-art method EERM (Wu et al., 2023) which is exclusively developed for graph OOD issues, and the test-time graph transformation state-of-the-art method GTRans (Jin et al., 2023). All methods are evaluated with four popular GNN backbones: GCN (Kipf & Welling, 2022), GraphSAGE (Hamilton et al., 2017), GAT (Veličković et al., 2018), and GPR (Chien et al., 2021). Their default setup follows that in EERM[1]. More implementation details of the baselines and GOAT can be found in Appendix

---

[1] Adjustments have only been made to the hidden dimensions of GAT to ensure consistency in the parameter count across all four backbones.

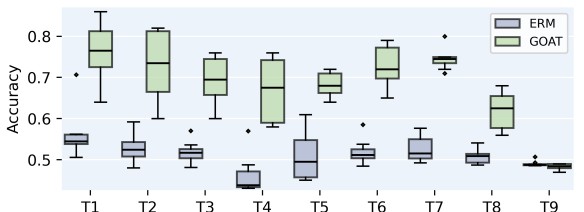

Figure 3: Results on Elliptic under OOD. GOAT improves SAGE on most test graphs.

Table 4: Efficiency comparison. GOAT is more time- and memory-efficient than EERM on large graphs and comparable to GTRANS.

|  | Running Time (s) | | | GPU Memory (GB) | | |
|---|---|---|---|---|---|---|
|  | Photo | Elliptic | ArXiv | Photo | Elliptic | ArXiv |
| EERM | 413.4 | 629.6 | - | 10.5 | 12.8 | 24+ |
| GTRANS | 1.9 | 6.8 | 12.2 | 1.6 | 1.3 | 4.1 |
| GOAT | 5.5 | 0.5 | 0.3 | 1.5 | 1.3 | 5.0 |

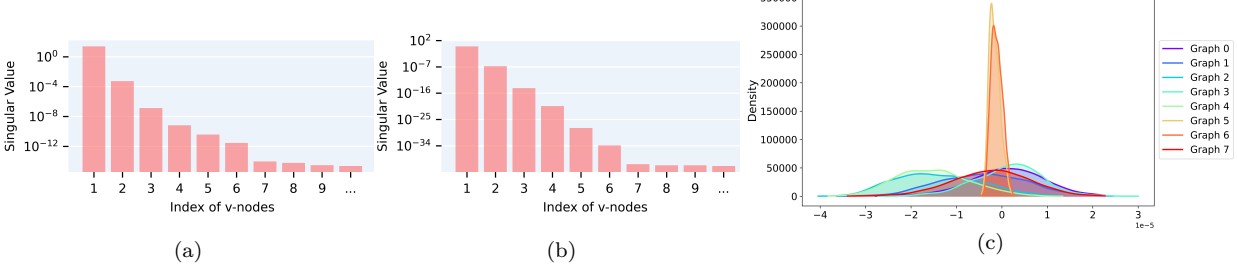

Figure 4: (a)(b) Visualization of the low-rank property of matrix $\Delta X$ in the LRA module of a GAT backbone trained on the two largest test graphs on OGB-ArXiv(169343 nodes) and FB-100(41554 nodes) under OOD settings. The singular values, obtained via SVD, show a rapid decay, indicating that **node embeddings can be effectively compressed into virtual nodes of the units digit**. (c) Visualization of the distribution of generated $\Delta X$ obtained after training GOAT on 8 test graphs in Cora with Gaussian KDE. The x-axis represents the sum of feature values on the nodes, and the y-axis represents the density of bias on each node within that value range. The further the mode of the distribution is from "0", the greater the degree of distribution shift.

C.1. All experiments are repeated 8 times with different random seeds. Due to page limits, additional baselines and backbones such as SR-GNN (Zhu et al., 2021), UDA-GCN (Wu et al., 2020), and GTN (Yun et al., 2019) are included in Appendix D.

**Overall Comparison.** Table 2 reports the averaged performance over the test graphs for each dataset as well as the average rank of each algorithm. From the table, we conduct the following observations: *(a) Overall Performance.* The proposed framework consistently achieves strong performance across the datasets: GOAT achieves average ranks of 1.3, 2.5, 2.3, and 1.7 with GCN, SAGE, GAT, and GPR, respectively, while the corresponding ranks for the best baseline GOAT are 1.8, 2.5, 2.3 and 2.4. Furthermore, in most cases, GOAT significantly improves the vanilla baseline (ERM) by a large margin. Particularly, when using SAGE as the backbone, GOAT outperforms ERM by 9.8%, 18.5%, and 3.9% on Cora, Elliptic, and OGB-ArXiv, respectively. These results demonstrate the effectiveness of GOAT in tackling diverse types of distribution shifts. *(b) Comparison to other baselines.* Both GraphCTA and EERM modify the model parameters to improve model generalization. Nonetheless, they are less effective than GOAT , as GOAT takes advantage of adapting the pre-trained GNN to the environment of test graphs. As test-time training methods, Tent and GTRANS also perform well in some cases. However, Tent is ineffective for models that do not incorporate batch normalization. On the other hand, GTRANS not only modifies node features but also alters edges, which can backfire if the edge modifications are not carefully chosen, potentially leading to a misrepresentation of the graph structure.

We further show the performance on each test graph on Elliptic with SAGE in Figure 3 and the results for other datasets are provided in Figure 9 in Appendix. We observe that GOAT generally improves over individual test graphs within each dataset, which validates the effectiveness of GOAT .

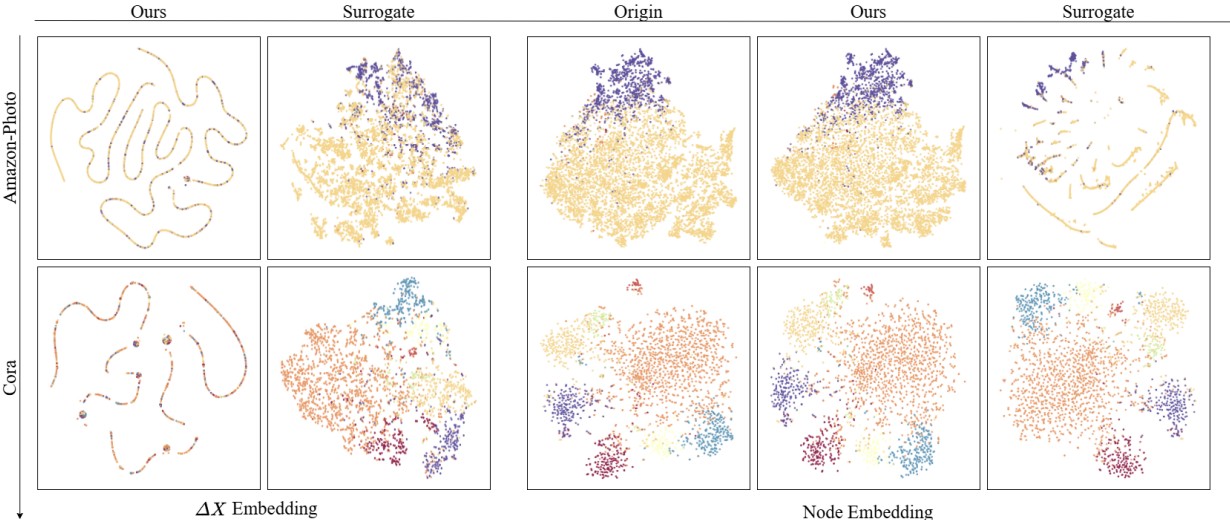

Figure 5: t-SNE visualizations of $\Delta X$ and node feature embeddings on Amazon-Photo, Cora, and datasets. The embeddings are obtained after a pre-trained GCN, using $\Delta X$ generated by our method and surrogate loss proposed by GTRANS.

**Efficiency Comparison.** In Table 4, we compare the computational time and GPU usage on the largest graph of each dataset for our GOAT , EERM, and GTRANS methods. Unlike EERM, which increases pre-training generalization through extensive environment augmentation during train time, GOAT optimizes efficiency by minimizing reliance on computationally expensive data augmentations and parameter tuning. In contrast to GTRANS, which adjusts based on the proportion of edges modified on the graph, GOAT requires sampling only a minimal number of *two* augmented input graphs per training epoch. These features ensure that GOAT not only conserves GPU resources but also accelerates the adaptation process during test time, showcasing substantial efficiency improvements over both train-time methods and other test-time methods.[2]

## 4.2 A Node-Level Low-Rank Perspective on Adaptation

After tuning the parameters of LRA on validation sets and obtaining the optimal results through test-time tuning, we further investigate the principal components of the low-rank matrix $\Delta X \in \mathbb{R}^{N \times d}$ in the $N$-dimensional space as Figure 4(a)(b) shown. By performing Singular Value Decomposition (SVD), we obtained the singular values sorted in descending order and compared the major eigenvalues, showing a significant decline compared to the others. In almost all large graphs, the adaptation graph $\Delta X$ exhibits low rank along its $N$-th dimension. This phenomenon differs from the low-rank attention applied to the node feature across each node's dimensions. This finding further demonstrates that the input test graph can be adapted using a low-rank additive representation. To elaborate, in Eq.(7), $\Delta X$ can also take the form $\Delta X \in \mathbb{R}^{N' \times d}$ with $N'$ nodes ($N' \ll N$), selected from a predefined node dictionary. This provides empirical evidence for setting our $|n|$ hyperparameter to a small constant that is independent of the number of nodes. In comparison, with full-node attention, the time cost will surge to 4x or even cause an OOM error.

## 4.3 An Embedding View on Adaptation $\Delta X$

We further visualize the t-SNE distribution of different $\Delta X$ generated by our paradigm and GTRANS on different test graphs. From Figure 5, it can be observed that the $\Delta X$ obtained through our paradigm, i.e., optimized with LRA module and $\mathcal{L}_{A2A}$, forms a low-dimensional manifold $\mathcal{M}$ after being embedded by the pre-trained GNN. This implies that $\Delta X$ itself belongs to a function manifold, encouraging $g_{\psi}(\cdot)$ to transform each node feature *along the local tangent space of $\mathcal{M}$ or along the curvature directions of $\mathcal{M}$.*

---

[2]Detailed early-stop procedures are shown in Appendix B.

Table 5: Ablation study of the overall loss function $\mathcal{L}_{A2A}$ comparison on the Elliptic dataset under OOD. Two-view sampling under a test environment shows improvement in GCN average performance on test graphs with the addition of each $\mathcal{L}_{A2A}$ constraint component, demonstrating the effectiveness of each part of the loss function and the choice of the number of samples.

| Configuration | | | | Performance | |
|---|---|---|---|---|---|
| Pre-train | $\mathcal{L}_s$ | $\mathcal{L}_c$ | $\mathcal{L}_R$ | One sample | Two samples |
| ✓ | | | | $\pm0.00\%$ | $\pm0.00\%$ |
| ✓ | ✓ | | | -1.51% | -1.51% |
| ✓ | ✓ | ✓ | | -1.62% | +4.49% |
| ✓ | ✓ | ✓ | ✓ | -0.18% | **+5.28%** |

Figure 6: $\alpha, \lambda$ Parameter Study. We compared the parameter sensitivity under two commonly used settings: $\alpha = 0.0005$ and $\lambda = 0.1$

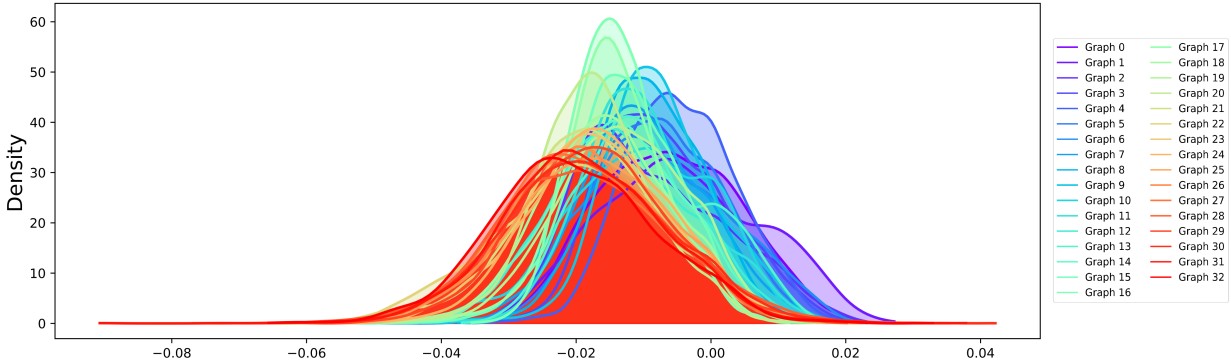

Figure 7: Visualization of the distribution of generated $\Delta X$ obtained after training GOAT on 32 test graphs in Elliptic with Gaussian KDE.

Furthermore, we found that although the $\Delta X$ generated by the GTRANS method still adheres to the clustering distribution of classification task labels, which, to some extent, even distorts the original topological structure learned by the GNN. A notable observation is that the performance of GTRANS correlates with the degree of distortion it introduces. The distortion, in effect, allows $\Delta X$ to generate a "spurious" graph that potentially overshadows the importance of the original node features for the classification task. Upon examining the node embeddings before and after transformation by $\Delta X$, it becomes evident that our method significantly enhances performance while maintaining a largely consistent distribution. This further demonstrates that the $\Delta X$ generated by our approach represents the degree of distributional shift, rather than conforming to the classification and clustering observed during pre-training.

## 4.4 Distribution Shift Quantification

We utilize Kernel Density Estimation (KDE) to visualize the distribution of $\Delta X$ generated by our adapter obtained through GOAT on Cora and Elliptic as Figure 4(c) and Figure 7 shown, by aggregating each node's feature dimensions $d$ in $\Delta X$. As the initialization of the $\Delta X$ is zero, the mean and mode of the initial distribution should be 0. Due to the varying degrees of distribution shifts in different graphs, after tuning by GOAT , our adapter can effectively capture the discrepancy between the current test graph and the pre-trained GNN. Adding the generated $\Delta X$ can be seen as the mapping from the current test graph to the distribution to which the original training graph belongs. Therefore, the farther the representation deviates from the origin, the more severe distribution shift the graph has, whether observed from the perspective of the entire graph or an individual node's perspective. The distribution shifts in the time-evolving graph could be more intuitive as time flows. Furthermore, we show $\Delta X$'s distribution in other datasets and compare with the central moment discrepancy (CMD) (Zellinger et al., 2022) measurement in the Appendix.Figure 10, highlighting the interpretability of our designed adapter.

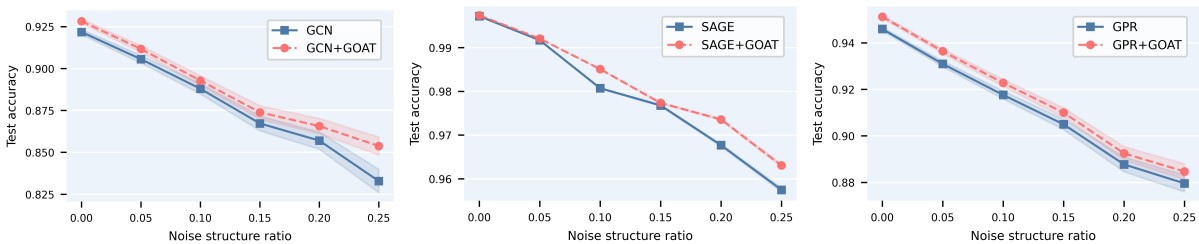

Figure 8: Performance of GCN, GraphSAGE, and GPR on Cora under structural attacks (edge addition/deletion).

### 4.5 Ablation Studies and Parameter Study

**Overall Loss Function $\mathcal{L}_{A2A}$.** The ablation study shown in Table 5 demonstrates the effectiveness of the different components in our proposed loss function. Optimizing $\mathcal{L}_s$ alone may lead to instability and mode collapse, which empirically proves our proposed target in Eq.(5) and Eq.(12). The target $\mathcal{L}_R$ is affected by whether the pre-trained GNN has sufficient generalization capability. Therefore, optimizing $\mathcal{L}_c$ alone is also sufficient to meet the requirements of Eq.(5). The best performance was achieved when all three components were jointly optimized. Additionally, sampling at least two distinct graph augmentations in $\widetilde{D}_{\mathrm{aug}}$ is essential to prevent biased learning, which empirically supports our choice of enforcing $(q \neq p)$ in Eq.(5). We show a more detailed loss component ablation study with backbone GraphSAGE in Appendix.E.1.

**Adapter LRA.** In Figure 6, we show the parameter study of $\lambda$ and $\alpha$ in $\mathcal{L}_{A2A}$. Note that while the proportion $\frac{\lambda}{1-\alpha}$ may vary, $1 - \alpha$ should remain relatively large to ensure that the constraint in Eq. (11), i.e., $\mathcal{L}_R$, is satisfied before optimizing the other objectives.

**Number of sampled views.** We further show the time cost and the performance in more sampled views ($\geq 2$) in Appendix.E.4. The results validate our choice that using two samples strikes an excellent balance, capturing the vast majority of the potential performance gain while maintaining the high efficiency that is critical for a practical test-time adaptation method.

### 4.6 Robustness under Structure Adversarial Attack

We evaluate the robustness of GCN, GraphSAGE, and GPR on the Cora dataset (Yang et al., 2016) under structural attacks, we perturb the graph by both adding and deleting edges. For a given perturbation budget ratio, we randomly delete a number of edges corresponding to ratio/2 of the original edge count, and randomly add an equal number of new edges. For the GCN model, our method achieves an average improvement of 2% under 25% structural attack noise, while for GPR, it delivers a stable gain of 0.5% across all noise levels. Without modifying hyperparameters (e.g., learning rate, sampling ratio), our method consistently improves backbone model performance by 0.5%–2.5% across varying attack intensities. Although our test-time paradigm consistently enhances adversarial robustness, its performance remains dependent on the base backbone's inherent generalization capability and adversarial robustness. Notably, this gain is achieved with just two samples with any subgraph sampling strategy, demonstrating computational efficiency alongside robustness.

### 4.7 Further Analysis

**Universal Bias vs. Local-global Bias.** We compare the average improvement of various non-customized additional parameter methods using our proposed $\mathcal{L}_{A2A}$ on graph classification datasets HIV (a small-scale real-world molecular dataset adapted from MoleculeNet (Wu et al., 2018)) under the OOD setting in GOOD (Gui et al., 2022) during test time. Average results of different designs of basis are shown in Table 6.

We show its improvement of ROC-AUC(%) based on ERM with GCN. For node classification, it is evident that UPF's universal prompts (Fang et al., 2024), $\Delta X \in \mathbb{R}^{1 \times d}$, across all nodes, are less customizable

for classifying each node in an OOD environment, and might even learn controversial knowledge, therefore showing diminishing performance.

Moreover, using a selection dictionary (Sun et al., 2023), $\Delta X \in \mathbb{R}^{k \times d}$ ($k \ll N$), also presents difficulties during test-time training. In contrast, subgraph-focused methods (Sun et al., 2022) can simultaneously capture the optimal bias more effectively, yielding relatively higher results, especially when it extends to a node-wise bias, i.e. each node's learnable bias is different, $\Delta X \in \mathbb{R}^{N \times d}$. These demonstrate that, at least in OOD node classification, a bias design that focuses on local-global context can better capture the relationships of nodes within the OOD environment. The node-

| Method | Avg. Impr |
|---|---|
| Universal | $\pm 0.01\%$ |
| Prompt dict | $+0.01\%$ |
| Sub-graph | $+1.52\%$ |
| Node-wise | $+2.35\%$ |

Table 6: Bias Comparison

wise bias method is particularly well-suited to our designed strategy, as it can better adapt to different distribution shift scenarios. This further validates the rationality of our adapter's design. Furthermore, based on the graph prompt methods (Liu et al., 2023; Yu et al., 2023), we experimented with incorporating a learnable scaling parameter that multiplies the weights of each GNN layer or node embeddings during test time. However, we found this approach difficult to apply effectively in our context.

## 5 Further Discussion

It is worth noting that the distribution shift issue in graph models can lead to significant risks and negative consequences in real-world applications. For instance, when GNNs are applied in financial risk control systems, distribution shifts in the input data may cause a large number of misjudgments, leading to severe economic losses or compliance issues. Similar risks exist in other high-stakes domains such as healthcare and cyber-criminal justice, where the reliability and robustness of graph-based decision-making systems under distributional changes are critical. Therefore, it is crucial to develop effective methods to detect and adapt to distribution shift scenarios in graph learning and to carefully analyze and mitigate the potential negative societal impacts. Our work aims to contribute to this important research direction.

Another illustrative example of the potential negative impact of the distribution shift issue in graph models is in the context of social network analysis for misinformation detection. GNNs have been widely adopted to identify fake news and rumors based on the propagation patterns and content features in social networks. However, the characteristics of misinformation can evolve rapidly over time, leading to distribution shifts between the training and test data. If the GNN-based misinformation detectors fail to adapt to such changes, they may miss emerging misinformation or cause false alarms, which can have severe societal consequences such as public panic, political manipulation, and erosion of trust in media. This urges the development of graph OOD detection and adaptation methods that can robustly handle the dynamic and adversarial nature of online misinformation. Our GOAT framework takes a step towards this goal by enabling test-time adaptation of GNNs to evolved data distributions.

## 6 Conclusion

In this work, we proposed **GOAT** , a data-centric and self-supervised test-time adaptation framework for graph neural networks that avoided the complexity and instability of structure-based transformations. By focusing exclusively on node feature optimization, GOAT enabled efficient and interpretable adaptation to distribution-shifted test graphs without requiring access to training data or test labels. To further enhance robustness, we introduced a low-rank adapter that generated diverse, graph-specific transformations and mitigated the risk of transformation collapse. Extensive experiments on six real-world datasets demonstrated that GOAT consistently improved performance across different GNN backbones, outperforming existing data-centric and model-centric test-time adaptation baselines. These results highlighted the potential of feature-only, data-centric approaches for test-time graph adaptation. In future work, we will explore a more relaxed optimization objective and inspire more discussion and novel propositions.

## Acknowledgments

This work was supported by the National Science Foundation (award number 2451605). Any opinions, findings, conclusions, or recommendations expressed in this material are those of the authors and do not necessarily reflect the views of the NSF. We thank Professor Tianhao Wang for critical feedback on the proofs and Tiange Lyu for valuable input during the early stages of this work.

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

# A Proof of Propositions

**Theorem 1.** *(Proved in GPF(Fang et al., 2024))* Given a pre-trained GNN model $f$, an input graph $\mathcal{G}$, for any graph-level transformation $g : \mathcal{G}\{\mathcal{A}, \mathcal{X}\} \to \mathcal{G}'\{\mathcal{A}', \mathcal{X}'\}$, there exists an additional extra feature vector $\Delta X$ that satisfies:

$$f(\mathcal{A}, \mathcal{X} + \Delta X) = f(g(\mathcal{A}, \mathcal{X})) = f(\mathcal{A}', \mathcal{X}')$$

## A.1 Population-Level Optimum of $\mathcal{L}_{A2A}$

We first characterize the theoretical optimum of the A2A loss in the population setting (infinite augmented views).

**Proposition A.1.** (Population Optimum) Suppose $f'_p = f(g_\psi(\mathcal{G}_p))$ is the adapted view representation and $f_q = f(\mathcal{G}_q)$ is the unadapted view representation, where $\mathcal{G}_p$ and $\mathcal{G}_q$ are independently and identically distributed (i.i.d.) samples from a stochastic augmentation process $Aug(\cdot)$.

The population $\mathcal{L}_{A2A}$ is defined as:

$$\mathcal{L}_s = \mathbb{E}_{p,q \sim Aug} \left[ \|f'_p - f_q\|^2 \right]$$

The optimal adapted representation that minimizes $\mathcal{L}_s$ is:

$$f'^*_p = \mathbb{E}_{q \sim Aug}[f_q] \triangleq \mu_{aug}$$

where $\mu_{aug}$ is a constant vector representing the expectation of all possible augmented view representations.

**Proof of Proposition A.1:**

$\mathcal{L}_s$ can be reformed as:

$$\mathcal{L}_s = \mathbb{E}_p \left[ \mathbb{E}_q[\|f'_p - f_q\|^2 \mid p] \right]$$

For any fixed $p$, the inner expectation $\mathbb{E}_q[\|f'_p - f_q\|^2 \mid p]$ is a standard mean squared error problem. By the property of expectation, $\arg\min_V \mathbb{E}[\|V - X\|^2] = \mathbb{E}[X]$, the optimal solution is:

$$f'^*_p = \mathbb{E}_q[f_q \mid p]$$

Since $\mathcal{G}_p$ and $\mathcal{G}_q$ are i.i.d. from $Aug(\cdot)$, the distribution of $f_q$ is independent of $p$:

$$f'^*_p = \mathbb{E}_q[f_q] = \mu_{aug}$$

This shows that the population optimum is a constant mapping for all inputs.

∎

**Remark 1.** Proposition A.1 reveals an important theoretical property: the unconstrained A2A loss drives all adapted representations toward the same constant $\mu_{aug}$. While this may appear similar to the *transformation collapse* we observe in practice, there is a crucial distinction: theoretical collapse converges to the correct constant $\mu_{aug}$, whereas practical collapse typically converges to a suboptimal constant that fails to capture the true augmentation distribution. Furthermore, while this is the theoretical optimum, our finite-sample algorithm with regularization operates in a different regime.

## A.2 Finite-Sample Behavior and Empirical Optimum

In practice, we work with a finite number of augmented views. This section analyzes the behavior of the empirical $\mathcal{L}_{A2A}$.

**Proposition A.2.** (Empirical Optimum) Given $n$ augmented views $\{\mathcal{G}_i\}_{i=1}^n$ at test time, let $f'_i = f(g_\psi(\mathcal{G}_i))$ be the adapted representations and $f_j = f(\mathcal{G}_j)$ be the fixed unadapted representations. The empirical $\mathcal{L}_{A2A}$ is:

$$\mathcal{L}_s^{emp} = \frac{1}{n(n-1)} \sum_{i=1}^{n} \sum_{j \neq i} \|f'_i - f_j\|^2$$

The empirical optimal solution that minimizes $\mathcal{L}_s^{emp}$ (treating $\{f_j\}$ as fixed) is the leave-one-out (LOO) mean:

$$f_i^{\prime emp} = \frac{1}{n-1} \sum_{j \neq i} f_j, \quad \forall i \in \{1, \ldots, n\}$$

**Proof of Proposition A.2:**

The empirical loss $\mathcal{L}_s^{emp}$ is a convex function that is separable in each $f_i'$. Taking the gradient with respect to $f_i'$ and setting it to zero:

$$\nabla_{f_i'} \mathcal{L}_s^{emp} = \nabla_{f_i'} \sum_{j \neq i} \|f_i' - f_j\|^2 = 0$$

$$\sum_{j \neq i} 2(f_i' - f_j) = 0$$

$$(n-1)f_i' = \sum_{j \neq i} f_j$$

$$f_i^{\prime emp} = \frac{1}{n-1} \sum_{j \neq i} f_j$$

∎

## A.3 Mean Equivalence of Empirical Optimum and Samples

**Proposition A.3.** The average of all empirical optimal adapted representations equals the sample mean of unadapted representations:

$$\bar{f}_n^{\prime emp} = \frac{1}{n} \sum_{i=1}^{n} f_i^{\prime emp} = \frac{1}{n} \sum_{j=1}^{n} f_j = \bar{f}_n$$

**Proof of Proposition A.3:**

Computing the average of empirical optima:

$$\bar{f}_n^{\prime emp} = \frac{1}{n} \sum_{i=1}^{n} f_i^{\prime emp} = \frac{1}{n} \sum_{i=1}^{n} \left( \frac{1}{n-1} \sum_{j \neq i} f_j \right)$$

For any fixed representation $f_j$, it appears in the inner sum for all $i \neq j$, i.e., it appears exactly $(n-1)$ times:

$$\bar{f}_n^{\prime emp} = \frac{1}{n(n-1)} \sum_{j=1}^{n} (n-1) f_j = \frac{1}{n} \sum_{j=1}^{n} f_j = \bar{f}_n$$

∎

**Remark 2.** (Key Insight on Avoiding Collapse)

Propositions A.2 and A.3 reveal why transformation collapse does not occur in practice despite the theoretical prediction from Proposition A.1:

1. Input-Dependent Diversity: Each $f_i^{\prime emp}$ is *different* for different $i$. Specifically:

$$f_i^{\prime emp} - f_k^{\prime emp} = \frac{1}{n-1}(f_k - f_i) \neq 0$$

The adapted representations remain input-dependent and diverse.

2. Finite-Sample Effect: Only when $n \to \infty$ do all $f_i^{\prime emp}$ converge to the same constant $\mu_{aug}$. For finite $n$ used in practice (e.g., $n = 2$ or $n = 4$), substantial diversity is preserved.

3. Regularization via Architecture: Our low-rank adapter parameterization $g_\psi$ further constrains the solution space, preventing the algorithm from reaching even the finite-sample optimum $\{f_i'^{emp}\}$.

Therefore, the practical algorithm operates in a regularized, finite-sample regime that naturally avoids the constant-mapping degeneracy predicted by population-level analysis.

### A.4 Convergence Analysis

We now analyze how the average adapted representation converges to the population mean as $n$ increases.

**Theorem 2.** (Convergence Rate via Bernstein's Inequality)

Let $\{\mathcal{G}_i\}_{i=1}^n$ be $n$ i.i.d. augmented views with representations $\{f_i = f(\mathcal{G}_i)\}_{i=1}^n$, $f_i \in \mathbb{R}^{N \times d}$. Define: $\mu_{aug} = \mathbb{E}[f_i]$ (population mean) and $\bar{f}_n = \frac{1}{n} \sum_{i=1}^n f_i$ (sample mean)

Assume: 1. Representations are bounded: $\|f_i\| \leq M$ almost surely 2. Variance bound: $\mathbb{E}[\|f_i - \mu_{aug}\|^2] \leq \sigma^2$

Then, applying the Vector Bernstein Inequality, for any $\delta \in (0, 1)$, with probability at least $1 - \delta$:

$$\|\bar{f}_n - \mu_{aug}\| \leq \sqrt{\frac{2\sigma^2 \log(2Nd/\delta)}{n}} + \frac{2M \log(2Nd/\delta)}{3n}$$

where $Nd$ is the node number times the dimension of representation $f_i$.

By Proposition A.3, since $\bar{f}_n'^{emp} = \bar{f}_n$, this bound also applies to the average of empirical adapted representations.

**Proof of Theorem 2:**

Define zero-mean random variables:

$$Z_i = f_i - \mu_{aug}, \quad i = 1, \ldots, n$$

By definition, $\mathbb{E}[Z_i] = 0$. We verify the conditions for vector Bernstein's inequality:

1. **Zero mean**: $\mathbb{E}[Z_i] = 0$ by construction.

2. **Bounded variance**:
$$\mathbb{E}[\|Z_i\|^2] = \mathbb{E}[\|f_i - \mu_{aug}\|^2] \leq \sigma^2$$

3. **Bounded range**: Since $\|f_i\| \leq M$ and $\|\mu_{aug}\| = \|\mathbb{E}[f_i]\| \leq M$ (by Jensen's inequality):

$$\|Z_i\| = \|f_i - \mu_{aug}\| \leq \|f_i\| + \|\mu_{aug}\| \leq 2M \triangleq B$$

Consider the sum $S_n = \sum_{i=1}^n Z_i$. By the vector Bernstein inequality, for any $t > 0$:

$$\mathbb{P}\left(\|S_n\| \geq t\right) \leq 2Nd \cdot \exp\left(-\frac{t^2/2}{n\sigma^2 + Bt/3}\right)$$

Setting the right-hand side equal to $\delta$ and solving for $t$:

$$2Nd \cdot \exp\left(-\frac{t^2/2}{n\sigma^2 + Bt/3}\right) = \delta$$

$$-\frac{t^2/2}{n\sigma^2 + Bt/3} = \log(\delta/2Nd)$$

$$\frac{t^2/2}{n\sigma^2 + Bt/3} = \log(2Nd/\delta)$$

This is a quadratic inequality in $t$. Solving yields:

$$t \leq \sqrt{2n\sigma^2 \log(2Nd/\delta)} + \frac{B\log(2Nd/\delta)}{3}$$

Since $\bar{f}_n - \mu_{aug} = \frac{1}{n}S_n$, dividing both sides by $n$:

$$\left\|\bar{f}_n - \mu_{aug}\right\| \leq \sqrt{\frac{2\sigma^2 \log(2Nd/\delta)}{n}} + \frac{B\log(2Nd/\delta)}{3n}$$

Substituting $B = 2M$ gives the stated bound.

∎

**Corollary 1.** (Convergence Rate)

The bound in Theorem 2 consists of two terms: Variance term: $O(1/\sqrt{n})$ (dominant for large $n$) and Bounded term: $O(1/n)$ (faster decay)

Therefore, as $n \to \infty$:

$$\left\|\bar{f}_n - \mu_{aug}\right\| = O\left(\frac{1}{\sqrt{n}}\right)$$

This establishes that the average adapted representation converges to the population mean $\mu_{aug}$ at the standard parametric rate.

**Remark 3.** (Practical Implications)

The convergence analysis provides several insights:

1. Sample Efficiency: The $O(1/\sqrt{n})$ rate means that doubling the number of views $n$ reduces the error by a factor of $\sqrt{2} \approx 1.41$. This suggests moderate values of $n$ (e.g., $n = 2$ to 4) already provide substantial benefits.

2. Diminishing Returns: Beyond a certain $n$, the computational cost of generating and processing additional views outweighs the marginal improvement in convergence, explaining our empirical observation that performance saturates.

3. Connection to Collapse: While $\bar{f}_n \to \mu_{aug}$ (collapse in the mean), individual $f_i'^{emp}$ maintain diversity through their leave-one-out structure. The regularization from our low-rank adapter further prevents collapse at the individual level.

## B    Algorithm

For detailed pseudocode under the augmented graphs number of $|\tau| = 2$, please refer to the right-hand side of the page.

---

**Algorithm 1 GOAT**  for Full Test-Time Graph Adaptation

---

1: **Input:** Pre-trained GNN $f_{\theta^\star}$ ($\theta^\star$ is fixed, without the last layer which is the classifier head) and test graph $\mathcal{G}_{te} = \{\mathcal{A}, \mathcal{X}\}$, Sample method(DropEdge) $Aug(\cdot)$
2: **Output:** Model prediction $\hat{Y}$
3: Initialize $\text{LRA}_\psi, \alpha, \lambda$, learning rate $\eta$
4: $\mathcal{L}_{\text{best}} = \infty$, patience = k, patience$_{\text{now}}$ = 0
5: **for** $t = 1$ to $T$ **do**

6:     $\mathcal{G}' = DE(\mathcal{G}_{te}) = \{\mathcal{A}', \mathcal{X}'\}$ , $\mathcal{G}'' = DE(\mathcal{G}_{te}) = \{\mathcal{A}'', \mathcal{X}''\}$

7:     $\Delta\hat{X}' = \text{LRA}_\psi(\mathcal{A}', \mathcal{X}')$, $\Delta\hat{X}'' = \text{LRA}_\psi(\mathcal{A}'', \mathcal{X}'')$

8:     $\Delta\hat{X}'_{emb} = f_{\theta^\star}(\mathcal{A}', \Delta\hat{X}')$ , $\Delta\hat{X}''_{emb} = f_{\theta^\star}(\mathcal{A}'', \Delta\hat{X}'')$

9:     $z'_{p_{emb}} = f_{\theta^\star}(\mathcal{A}', \mathcal{X}'+\Delta\hat{X}')$ , $z''_{p_{emb}} = f_{\theta^\star}(\mathcal{A}'', \mathcal{X}''+\Delta\hat{X}'')$

10:     $z'_{emb} = f_{\theta^\star}(\mathcal{A}',\mathcal{X}')$ , $z''_{emb} = f_{\theta^\star}(\mathcal{A}'', \mathcal{X}'')$

11:     $\mathcal{L}_s = \mathbb{E}\|(z'_{p_{emb}} - z''_{emb}) + (z''_{p_{emb}} - z'_{emb})\|^2$

12:     $\mathcal{L}_c = \mathbb{E}\|(z'_{p_{emb}} - z'_{emb} - \Delta\hat{X}'_{emb}) + (z''_{p_{emb}} - z''_{emb} - \Delta\hat{X}''_{emb})\|^2$

13:     $\mathcal{L}_R = \mathbb{E}|\Delta\hat{X}'_{emb} + \Delta\hat{X}''_{emb}|$

14:     $\mathcal{L} = \alpha\lambda(\mathcal{L}_s + \mathcal{L}_c) + (1 - \lambda)\mathcal{L}_R$

15:     **Update:** $\psi \leftarrow \psi - \eta\Delta_\psi L$
16:     **if** $\mathcal{L}_R \leq \mathcal{L}_{best}$ **then**
17:         $\mathcal{L}_{best} = \mathcal{L}_R$
18:         patience$_{\text{now}}$ = 0
19:     **else**
20:         patience$_{\text{now}}$ = patience$_{\text{now}}$ + 1
21:     **end if**
22:     **if** patience$_{\text{now}}$ ≥patience **then**
23:         **Stop**
24:     **end if**
25: **end for**
26: $\Delta X = \text{LRA}_\psi(\mathcal{A}, \mathcal{X})$
27: $\hat{Y} = f_{\theta^\star}(\mathcal{A}, \mathcal{X} + \Delta X)$
28: **return** $\hat{Y}$

---

## C    Datasets and Hyper-Parameters

In this section, we reveal the details of reproducing the results in the experiments. We will release the source code upon acceptance.

### C.1    Hyper-Parameter Setting

For the setup of backbone GNNs, we majorly followed EERM (Wu et al., 2023):

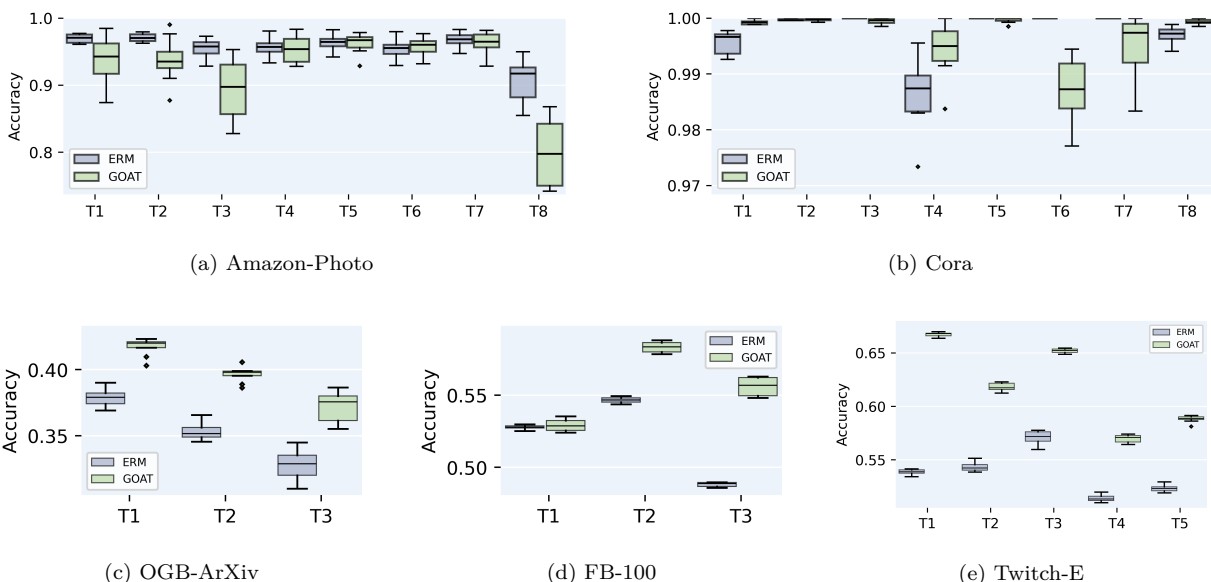

Figure 9: Classification performance on individual test graphs within each dataset for OOD setting.

(a) **GCN**: the architecture setup is 5 layers with 32 hidden units for Elliptic and OGB-ArXiv, and 2 layers with 32 hidden units for other datasets, with batch normalization for all datasets. The pre-train learning rate is set to 0.001 for Cora and Amz-Photo, 0.01 for other datasets; the weight decay is set to 0 for Elliptic and OGB-ArXiv, and 0.001 for other datasets.

(b) **GraphSAGE**: the architecture setup is 5 layers with 32 hidden units for Elliptic and OGB-ArXiv and 2 layers with 32 hidden units for other datasets, and with batch normalization for all datasets. The pre-train learning rate is set to 0.01 for all datasets; the weight decay is set to 0 for Elliptic and OGB-ArXiv, and 0.001 for other datasets.

(c) **GAT**: the architecture setup is 5 layers for Elliptic and OGB-ArXiv, and 2 layers for other datasets, with batch normalization for all datasets. Each layer contains 4 attention heads and each head is associated with 8 hidden units. The pre-train learning rate is set to 0.01 for all datasets; the weight decay is set to 0 for Elliptic and OGB-ArXiv, and 0.001 for other datasets.

(d) **GPR**: We use 10 propagation layers and 2 transformation layers with 32 hidden units. The pre-train learning rate is set to 0.01 for all datasets; the weight decay is set to 0 for Elliptic and OGB-ArXiv, and 0.001 for other datasets. Note that **GPR does not contain batch normalization layers**.

For the baseline methods, we tuned their hyper-parameters based on the validation performance. For Tent, we search the learning rate in the range of [1e-2, 1e-3, 1e-4, 1e-5] and the running epochs in [1, 10, 20, 30]. For EERM(Wu et al., 2023) and GTʀᴀɴs(Jin et al., 2023), we followed the instructions provided by the original paper. For GraphCTA(GCTA)(Zhang et al., 2024), we tune the feature adaptation $\eta_1$ in [5e-3, 1e-3, 1e-4, 1e-5, 1e-6], learning rate of structure adaptation $\eta_2$ in [0.5, 0.1, 0.01], and alternatively optimize node features epochs $\tau_1$ in [1, 2, 3] and optimize graph structure epochs $\tau_2$ in [1, 2], other parameters followed the instruction provided by the original paper. For GOAT , we adopt *DropEdge* as the augmentation function $\mathcal{DE}(\cdot)$ and set the drop ratio to 0.05, K-layer aggregation in LRA set to 1 due to some GNN only has two layers in some datasets while the last GNN layer performs as a classifier head. We use Adam Optimizer for LRA module tuning. We further search the learning rate $\eta$ in [1e-2, 5e-3, 1e-3, 5e-4, 1e-4, 5e-5, 1e-5, 1e-6] for different backbones, the virtual nodes number $|n|$ in [1×, 2×, 5×, 10×, 20×] of the class number $C$, the attention dim $d_{attn}$ in LRA in [2, 4, 8, 16, 32], total epochs $T$ in [50, 100], and the patience in [1, 0.5, 0.1, 5e-2, 1e-2, 1e-3]. In the optimization target, we search the $\lambda$ in [1, 3, 5, 10] and the $\alpha$ in [0.999, 0.9, 0.75,

| Method | Amz-Photo | Cora | Elliptic | FB-100 | OGB-ArXiv | Twitch-E |
|---|---|---|---|---|---|---|
| **ERM** | 93.79±0.97 | 91.59±1.44 | 50.90±1.51 | 54.04±0.94 | 38.59±1.35 | 59.89±0.50 |
| **UDA-GCN** | 91.70±0.35 | 92.65±0.46 | 51.57±1.31 | 54.11±0.54 | 39.43±0.71 | 52.12±0.38 |
| **SRGNN** | **94.64±0.17** | 94.08±0.28 | 51.94±0.81 | 54.08±1.10 | 38.92±0.65 | 59.21±0.51 |
| **GOAT** | 94.35±1.32 | **94.79±1.36** | **55.83±3.81** | **54.19±2.04** | **39.44±2.02** | **60.15±1.30** |

Table 7: Performance comparison between GOAT with GCN and graph domain adaptation methods.

| Method | Amz-Photo | Cora | Elliptic | FB-100 | OGB-ArXiv | Twitch-E |
|---|---|---|---|---|---|---|
| GTN | 94.73±2.91 | 99.88±0.10 | 68.51±3.85 | 53.57±0.75 | 43.08±0.84 | 62.30±0.16 |
| GTN + GOAT | 94.75±2.97 | 99.85±0.12 | 70.08±2.50 | 54.94±0.61 | 44.11±0.84 | 63.79±0.27 |

Table 8: Performance comparison between GOAT with other backbones.

0.5, 0.25, 0.1, 5e-2, 1e-2, 5e-3]. We note that the process of tuning hyper-parameters is quick due to the high efficiency of test-time adaptation as we demonstrated in Section 4.1. Furthermore, not every test graph is learned over whole epochs set due to the patience of dissatisfaction of constraint in Eq 12.

**Evaluation Protocol.** For ERM (standard pre-training), we pre-train all the GNN backbones using the common cross-entropy loss. For EERM, it optimizes a bi-level problem to obtain a trained classifier. Note that the aforementioned two methods do not perform any test-time adaptation and their model parameters are fixed during the test. For the four test-time adaptation methods, Tent, GCTA, GTRANS, and GOAT. We first obtain the GNN backbones pre-trained from ERM and adapt the model parameters or graph data at test time, respectively. Furthermore, Tent minimizes the entropy loss and GTRANS and GCTA both minimize the contrastive surrogate loss, while GOAT minimizes the Target $\mathcal{L}_{A2A}$.

## C.2 Hardware and Software Configurations

We perform experiments on NVIDIA GeForce RTX 3090 GPUs. The GPU memory and running time reported in Table 3 are measured on one single RTX 3090 GPU. Additionally, we use eight CPUs, with the model name Intel(R) Xeon(R) Silver 4210R CPU @ 2.40GHz. The operating system utilized in our experiments was Ubuntu 22.04.3 LTS (codename jammy).

# D More Experimental Results

## D.1 Overall Comparison

To show the performance on individual test graphs, we choose SAGE as the backbone model and include the box plot on all test graphs within each dataset in Figure 9. We observe that GOAT generally improves over each test graph within each dataset, which validates the effectiveness of our proposed method.

## D.2 Comparison to More Baseline and Backbones

To compare their empirical performance, we include two GraphDA methods (Zhu et al., 2021; Wu et al., 2020). SR-GNN regularizes the model's performance on the source and target domains. Note that SR-GNN was originally developed under the transductive setting where the training graph and test graph are the same. To apply SR-GNN in our OOD setting, we assume the test graph is available during the training stage of SR-GNN, as typically done in domain adaptation methods. UDA-GCN is another work that tackles graph data domain adaptation, which exploits local and global information for different domains. We followed the authors' suggestions in their paper to tune the hyper-parameters and the results are shown in Table 7. On the one hand, we can observe that these graph domain adaptation methods generally improve the performance of GCN under distribution shift and SRGNN is the best-performing baseline. On the other hand, GOAT performs the best on all datasets except Amz-Photo. On Amz-Photo, GOAT does not improve as much as SR-GNN, which indicates that joint optimization over source and target is necessary for this dataset.

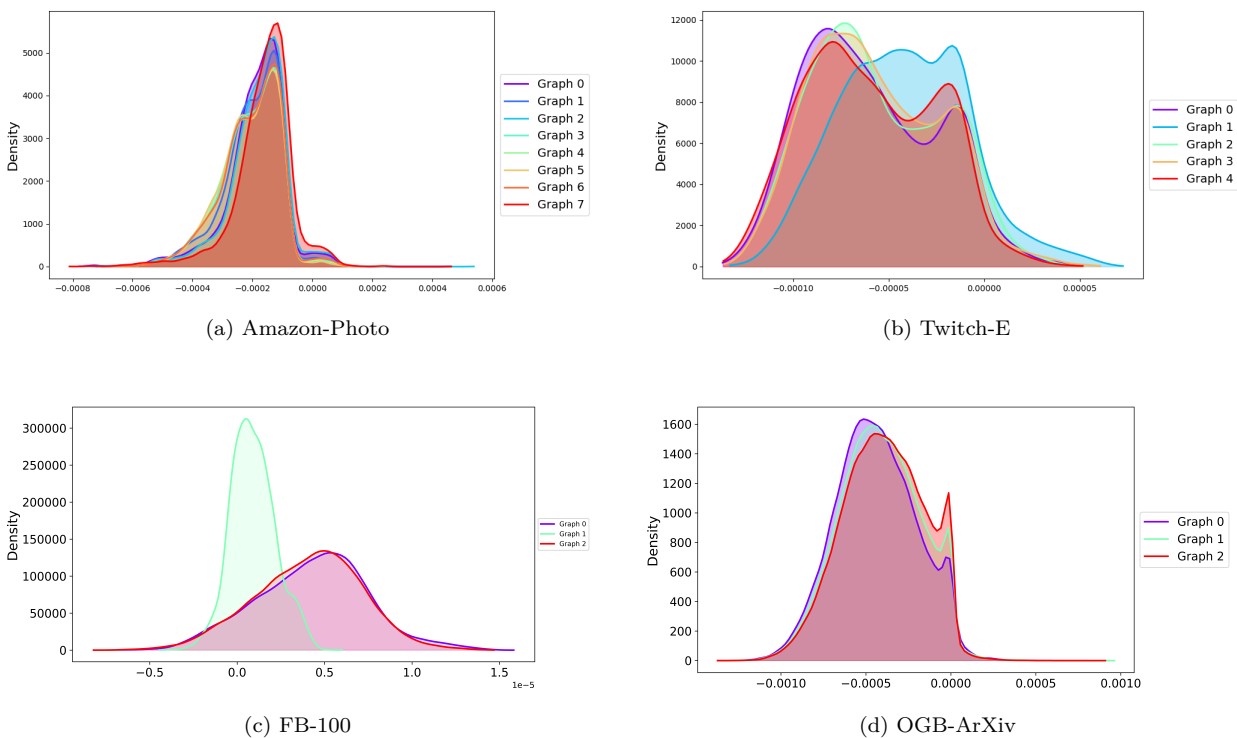

Figure 10: $\Delta X$ Distribution after training on each dataset.

| GraphID | $\mathcal{G}_0$ | $\mathcal{G}_1$ | $\mathcal{G}_2$ | $\mathcal{G}_3$ | $\mathcal{G}_4$ | $\mathcal{G}_5$ | $\mathcal{G}_6$ | $\mathcal{G}_7$ | $\mathcal{G}_8$ |
|---|---|---|---|---|---|---|---|---|---|
| Amz-Photo | 6.4 | 5.1 | 5.5 | 3.7 | 2.8 | 3.7 | 3.9 | 6.6 | - |
| Cora | 5.4 | 4.2 | 4.8 | 6.3 | 5.5 | 4.8 | 4.6 | 5.4 | - |
| Elliptic | 80.2 | 90.8 | 114.3 | 86.5 | 789.3 | 781.6 | 99.4 | 100.4 | 150.6 |
| OGB-ArXiv | 14.7 | 20.6 | 10.4 | - | - | - | - | - | - |
| FB-100 | 29.7 | 16.9 | 32.9 | - | - | - | - | - | - |
| Twitch-E | 8.6 | 6.1 | 9.0 | 8.4 | 9.7 | - | - | - | - |

Table 9: CMD values on each individual graph based on the pre-trained GCN.

However, recall that domain adaptation methods are less efficient due to the joint optimization on source and target. Overall, the test-time graph adaptation with our adapter could better fit the specific distribution shifts that deviate from the source target. As shown in Table 8, GOAT could also adapt to more popular backbones.

## D.3 Quantifying distribution shift through LRA

In this section, we further show more distribution of $\Delta X$ generated by GOAT that indicates the OOD degree of each test graph. We can see an extreme shift in Figure 10(c) that as the snapshots flow from the validation, the mode and the mean value of $\Delta X$ shift away from the 0 initialized value, which shows a further deviation of the later test graphs from the train source distribution. Furthermore, following SR-GNN (Zhu et al., 2021), we adopt central moment discrepancy (CMD) (Zellinger et al., 2022) as the measurement to quantify the distribution shifts in different graphs, we present them in Table 9 as a comparison with $\Delta X$ in GOAT .

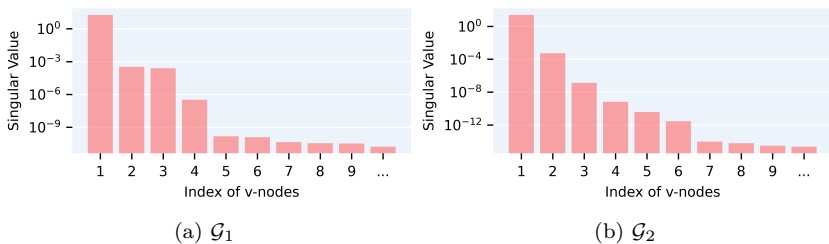

(a) $\mathcal{G}_1$            (b) $\mathcal{G}_2$

Figure 11: (a)(b) SVD of E in LRA after training on test graph $\mathcal{G}_1$, $\mathcal{G}_2$ in OGB-ArXiv with distribution shifts.

Table 10: Adaptation efficiency under different numbers of virtual nodes on Twitch-E ($C = 2$).

| # Virtual Nodes | Value | Time (s) | GPU Memory (GB) |
|---|---|---|---|
| Default Setting | $2C = 4$ | $1.50 \pm 0.38$ | 0.9 |
| | $4C = 8$ | $1.44 \pm 0.40$ | 0.9 |
| | $20C = 40$ | $1.54 \pm 0.41$ | 1.0 |
| | $200C = 400$ | $1.62 \pm 0.42$ | 1.1 |
| Full Graph (No Compression) | $N \approx 4000$ | $6.72 \pm 2.77$ | 3.4 |

### D.4 Low-rank of Node-level Representation on Large Graph

In Figure 11, we show other $\Delta X$ generated from LRA on two large test graphs in OGB-arXiv with 69499 and 120740 nodes.

### D.5 Efficiency of LRA design: Impact of the Number of Virtual Nodes

The number of virtual nodes, denoted as $r$, is a crucial hyperparameter in our low-rank cross-attention adapter, controlling the rank and expressive capacity of the transformation. In our main experiments, we set $r = 2C$ (where $C$ is the number of classes) as the default, based on a balance between efficiency and effectiveness. To evaluate the sensitivity of our method to this choice, we conduct an ablation study on the Twitch-E dataset, varying $r$ from $2C$ to $200C$, and compare against a full-graph baseline where adaptation is performed over all $N \approx 4000$ original nodes (denoted as "$N$ (Full Graph)").

The results are summarized in Table 10, reporting adaptation time per graph (in seconds) and overall GPU memory consumption during inference. All configurations achieve nearly identical performance, with final accuracy stabilized at **0.601**, indicating that the model's predictive quality is robust to the number of virtual nodes.

As shown in the table, increasing the number of virtual nodes has minimal impact on adaptation time until $r = 200C$, while memory usage grows only slightly due to the lightweight attention mechanism. In contrast, the full-graph baseline suffers from a **4.5× slower adaptation speed** and consumes **3.8× more GPU memory**, highlighting the effectiveness of our low-rank design.

Notably, even the smallest configuration ($r = 2C$) achieves the same performance as the full-graph model, demonstrating that our virtual nodes efficiently capture the essential transformation signals without requiring high-rank or full-node computation. This validates our design choice: using $r = 2C$ provides sufficient expressiveness while maintaining high efficiency, making it a scalable and practical default across datasets with varying sizes and label complexities.

# E   Ablation study

## E.1   Optimization Object

In Figure 11, we show the parameter study of $\lambda$ and $\alpha$ in $\mathcal{L}_{A2A}$. Noted that there could be a different proportion of $\frac{\lambda}{(1-\alpha)}$, while it still should have a rather value of $\alpha$ in that the constraint in Eq.(11) ($\mathcal{L}_R$) should be satisfied first then the other objective could work.

To evaluate the individual and combined contributions of each component of $\mathcal{L}_{A2A}$, we conduct an ablation study across all six datasets using GraphSAGE as the backbone. We report the average performance improvement (in percentage points) over the no-adaptation baseline. The results are summarized in Table 11.

Table 11: Ablation study on the components of the adaptation loss. All configurations include the base adapter ($\mathcal{L}_{A2A}$). Performance is measured as average improvement (%) over the no-adaptation baseline across six datasets with GraphSAGE.

| Method | Loss Components | | | Average Improvement (%) | |
|---|---|---|---|---|---|
| | $\mathcal{L}_s$ | $\mathcal{L}_c$ | $\mathcal{L}_R$ | Two-Sample | One-Sample |
| Baseline | | | | +0.00 | +0.00 |
| $+ \mathcal{L}_s$ | ✓ | | | +2.08 | +2.08 |
| $+ \mathcal{L}_c$ | | ✓ | | +2.15 | +2.09 |
| $+ \mathcal{L}_R$ | | | ✓ | +2.14 | +2.14 |
| $+ \mathcal{L}_s + \mathcal{L}_c$ | ✓ | ✓ | | +2.31 | +2.20 |
| $+ \mathcal{L}_s + \mathcal{L}_R$ | ✓ | | ✓ | +2.14 | +2.14 |
| $+ \mathcal{L}_c + \mathcal{L}_R$ | | ✓ | ✓ | +2.14 | +2.10 |
| Full $\mathcal{L}_{a2a}$ | ✓ | ✓ | ✓ | **+2.32** | +2.16 |

Key observations from the results include: Each individual loss term provides a consistent gain over the baseline, with $\mathcal{L}_c$ yielding the highest single-component improvement (+2.15%), followed closely by $\mathcal{L}_R$ (+2.14%) and $\mathcal{L}_s$ (+2.08%). The combination of $\mathcal{L}_s$ and $\mathcal{L}_c$ achieves the best overall performance (+2.31%), indicating a synergistic effect: the consistency constraint helps align the adapter with the GNN's inductive bias, while symmetry enforces view invariance. Adding $\mathcal{L}_R$ to other combinations does not further improve performance and can slightly degrade results when combined with $\mathcal{L}_s$, suggesting that excessive regularization may suppress meaningful adaptation signals. The full model (all three terms) achieves the highest two-sample improvement (+2.32%), validating the design of our decomposed objective.

These findings reveal that the components are not simply additive but exhibit a hierarchical and interdependent relationship. In particular, $\mathcal{L}_c$ plays a central role in ensuring structural compatibility, while $\mathcal{L}_R$ stabilizes training without dominating the update direction.

It is worth noting that the relative effectiveness of $\mathcal{L}_R$ appears somewhat weaker in this ablation (with SAGE) compared to the results on GCN backbones presented in the main paper. While the trend of $\mathcal{L}_c$ being the most impactful component remains consistent, the performance gain from $\mathcal{L}_R$ is less pronounced here. This discrepancy may stem from differences in the architectural sensitivity of GCN and SAGE to regularization, or from variations in message-passing dynamics under different aggregation schemes.

Nevertheless, the core insight—that $\mathcal{L}_c$ and $\mathcal{L}_R$ act as complementary mechanisms to stabilize and enable $\mathcal{L}_s$ rather than merely augment it—holds across architectures. This ablation confirms that our decomposition effectively balances expressiveness and optimizability, enabling robust and effective test-time adaptation, even if the exact contribution magnitudes vary across model families.

## E.2   Different Augmentation Methods Used in Optimization

In Target 4, we used *DropEdge* as the augmentation function $Aug(\cdot)$ to obtain the augmented view and enough inductive graphs. In practice, the choice of augmentation can be flexible, and here we explore two other choices: node dropping and *FlipEdge* (You et al., 2020). Specifically, we adopt a ratio of 0.05 for node

dropping, a ratio of 0.05 and 0.5 for *FlipEdge*, and ratios of 0.05 and 0.5 for *DropEdge*. We observe that (1) GOAT with any of the three augmentations can greatly improve the performance of GCN under distribution shift, and (2) different augmentations lead to slightly different performances on different datasets

### E.3 Parameters in LRA

After tuning over all datasets, the hyperparameter almost shows a slight difference. Therefore, $d_{attn}$ is set to 8, $|n|$ is almost $10 \times C$ ($C$ is the class number of nodes in test graph). Furthermore, we explore that, unlike the Transformers structure in NLP, the multi-head attention and the residual connection cannot improve the performance in our LRA module, which indicates the graph structure data information learned with GNNs has a different representation from that learned as in NLP as sequences.

### E.4 Choice of Number of Augmented Views ($K$)

The number of augmented views, denoted as $K$, is a key hyperparameter in our test-time adaptation framework. In the main paper, we adopt $K = 2$ as the default setting for all experiments. This choice is motivated by both robustness and efficiency considerations. Below, we provide a detailed ablation study to justify this design decision.

Using two augmented views provides a simple and stable baseline that ensures consistent hyperparameter settings across diverse datasets and backbone architectures. More importantly, our extended experiments demonstrate that while increasing $K > 2$ can yield marginal performance improvements, the gains quickly diminish relative to the significantly increased computational cost.

To illustrate this trade-off, we conduct ablation experiments on three representative datasets—Cora-GAT, Elliptic-GCN, and Twitch-E-GCN—with $K$ ranging from 2 to 20. The results are summarized in Table 12. We report both adaptation time per graph (in seconds) and final model performance.

Table 12: Performance and adaptation time under different numbers of augmented views ($K$).

| $K$ (# Views) | Cora-GAT | | Elliptic-GCN | | Twitch-E-GCN | |
|---|---|---|---|---|---|---|
| | Time (s) | Acc. | Time (s) | F1. | Time (s) | ROC-AUC. |
| 2 | 1.3689 | 0.9250 | 1.9075 | 0.5222 | 0.8734 | 0.6210 |
| 3 | 1.9199 | 0.9249 | 2.7447 | 0.5300 | 1.2444 | 0.6210 |
| 4 | 2.7233 | 0.9245 | 3.7554 | 0.5344 | 1.5541 | 0.6209 |
| 5 | 3.3211 | 0.9266 | 4.0833 | 0.5378 | 2.0369 | 0.6209 |
| 10 | 7.4531 | 0.9281 | 7.5256 | 0.5578 | 4.1890 | 0.6209 |
| 20 | 14.0208 | 0.9308 | 16.0332 | 0.5678 | 8.7940 | 0.6209 |

As shown in the table, the adaptation time increases substantially with $K$. This is due to two factors: (1) a linear increase in the number of GNN forward passes (one per view), and (2) a combinatorial increase in the number of pairwise similarity comparisons in the symmetry loss $\mathcal{L}_s$, which scales as $\binom{K}{2} = K(K-1)/2$.

The performance gains from using more than two views are marginal compared to the rising computational cost: on Cora-GAT, increasing $K$ from 2 to 10 yields only $+0.31\%$ improvement with a $5.4\times$ time increase; on Elliptic-GCN, going from $K = 2$ to $K = 5$ improves performance by $+1.56\%$, far less than the $+5.28\%$ gain from adding the second view, yet nearly doubles the cost; and on Twitch-E, no gain is observed beyond $K = 2$, indicating view diversity saturation.

These results confirm that the performance benefit of using more than two views is limited, especially when considering the super-linear growth in computational overhead. In contrast, $K = 2$ captures the vast majority of the potential gain from contrastive adaptation while maintaining high efficiency—making it an ideal choice for practical test-time adaptation. Therefore, we conclude that $K = 2$ offers an excellent balance between effectiveness and efficiency, and is well-suited as a general-purpose setting across various graph structures and model backbones.

