# OpenReview forum: "Avoiding Structural Pitfalls: Self-Supervised Low-Rank Feature Tuning for Graph Test-Time Adaptation"
_TMLR — Accepted by TMLR_

### Review · Reviewer_WHEa · 2025-07-26

**Summary Of Contributions:**

The summary of the contributions of the paper is as follows:
This paper introduces GOAT (Graph Optimization via Augmented Transformations), a novel self-supervised test-time tuning paradigm designed to adapt pre-trained graph neural networks (GNNs) to distribution-shifted test data by exclusively leveraging node feature transformations, thereby avoiding the limitations and instability of structure-based methods. To mitigate the issue of transformation collapse, the authors propose a parameter-efficient low-rank adapter that generates diverse, graph-specific transformations, enhancing both adaptation performance and interpretability without altering the underlying graph structure. Extensive experiments on six real-world datasets demonstrate that GOAT consistently improves the performance of various pre-trained GNN backbones, achieving state-of-the-art results under diverse distribution shifts.

**Audience:**

Yes

**Broader Impact Concerns:**

There are no "Broader Impact Concerns" for this paper.

**Claims And Evidence:**

Yes

**Requested Changes:**

Based on the aforementioned strengths and weaknesses, the requested changes are provided in an orderly manner from the highest to the lowest importance as follows:

- There are unreferenced terms such as mapping collapse or transformation collapse. If new to the audience, they must be clearly defined. If known to the community, references and elaborations to the concepts behind these terms are insightful to let the audience get connected to the related literature.

- the details on Fig. 1 right side for the test-time inference phase is missing.

- Elaboration on the effect of the 2 view choice rather than K view choice is insightful.

- Some empirical analysis on the choice of the layer k at which the features for the low-rank adaptor is used is insightful.

**Strengths And Weaknesses:**

# Strengths And Weaknesses

## Strengths
* A major strength is the method's exclusive focus on node feature transformations rather than graph structure changes, which simplifies the adaptation process and avoids the instability typically associated with structural modifications.

* The proposed self-supervised learning strategy is well-designed, leveraging embedding discrepancies between augmented graphs to learn distribution-aware feature transformations, which is both elegant and effective.

* The introduction of a parameter-efficient low-rank adapter to prevent transformation collapse is a valuable contribution. It ensures that each input graph receives a unique, meaningful transformation while preserving the knowledge from the pre-trained model.

* By avoiding graph structure changes and instead focusing on node-level adaptations, the method also enhances interpretability—making it easier to understand the effect of transformations.

* The experimental results are robust and compelling, showing consistent improvements across diverse datasets and backbone architectures, which highlights the generality and practical relevance of the approach.

- The attempt to provide a thorough emprical analysis of various aspects of the propsed method such as the efficieny, interpretability, robustness is very important and attentive.

## Weakness

- The major weakness is the quantitative results presented in table 3 where there are significant number of cases the performance of GOAT falls behind the SoTA. A lack of analysis and explanantion is obvious. Insights and narration of such cases is important for the reader to observe the pros and cons of the latest approaches in an unbiased manner.

- what is the transformation function g  in eq. 5 since the low-rank adaptation is not mentioned yet. It seems g(G) is simply adding the low-rank features to the augmented views. Please elaborate on this aspect and clarify. At the point the audience reaches eq.5 there is no clear view on what g is.

---

> ### Author Response · Authors · 2025-08-16
> **Response to Reviewer WHEa part I**
>
> We are very grateful to Reviewer WHEa for the exceptionally thorough and positive review. We are encouraged by the reviewer's appreciation for our method's design, novelty, and the rigor of our empirical analysis. We address the identified weaknesses and requested changes below.
>
> 1. **Regarding Weakness 1: Quantitative results in Table 2 (mistakenly referenced as Table 3).**
>
>     While GOAT is not the top performer in 100% of cases, it demonstrates remarkable consistency and robustness, which is reflected in its **superior average rank**. As shown in the "Avg." rank column in Table 2, GOAT achieves average ranks of 1.3, 2.5, 2.3, and 1.7 across the four backbones, consistently outperforming all baselines on average.
>
>     Our experimental observations indicate that the suboptimal performance on certain datasets is often correlated with the insufficient optimization of the third component of our loss function, $\mathcal{L}_R$, which represents the norm of the node embeddings after adaptation.
>
>     For instance, in the GPR-Elliptic case, the average value for $\mathcal{L}_R$ across the test set after optimization is approximately 0.12. In contrast, for most other datasets and backbone combinations where our method performs well, this value is an order of magnitude lower, typically 0.01 or less. We hypothesize this is a consequence of using unified hyperparameters across diverse settings. We also experimented with substituting our loss with GTrans's objective while using our LRA and sampling paradigm; the performance of this hybrid model surpassed our original design only when its corresponding $\mathcal{L}_R$ value was also minimized to a low level.
>
>     To elaborate from our loss function's design perspective, the minimization of $\mathcal{L}_R$ is foundational, as its optimization is crucial for the entire framework to hold. However, as a joint optimization problem, it is possible that on certain extreme datasets, the optimization of the other terms ($\mathcal{L}_s$ and $\mathcal{L}_c$) occurs at the expense of $\mathcal{L}_R$. This trade-off prevents the adaptation from successfully learning a low-dimensional manifold to represent the distribution shift. This issue can, however, be mitigated to some extent through more meticulous, case-specific hyperparameter tuning. Furthermore, as discussed in `Regarding Requested Change 4`, our method can reach better performance with a larger `K`. The performance shown in Table 2 is under a superior time efficiency with acceptable performance.
>
> 2. **Regarding Weakness 2 and Requested Change 2: Clarifying the transformation function $g$ in Eq. 5.**
>
>     This is an excellent point regarding the clarity and flow of our methodology section. However, in Section 3.1, we treat $g_\psi$ as any graph transformation, and then in Section 3.2, we find that there is a better solution, $g_\psi$ can only transform on the graph node feature.
>
> 3. **Regarding Requested Change 1: Defining "mapping collapse" or "transformation collapse".**
>
>     Thank you for this suggestion. We agree that this term should be defined more explicitly for the reader. We use "transformation collapse" to describe a failure mode where the learned feature transformation converges to a trivial or ineffective solution (e.g., an identity or zero mapping) for all inputs, thus failing to facilitate meaningful adaptation.
>
> - **Action:** In the revised manuscript, we will add a clear and concise definition of **"transformation collapse"** —— feature transformation fall into an identity or zero mapping in **Section 3.2**, the first time it is mentioned in both the **abstract and the introduction**.
>
> (more response in the next)

---

> ### Author Response · Authors · 2025-08-16
> **Response to Reviewer WHEa part II**
>
> 4. **Regarding Requested Change 2: Details on Fig. 1 (test-time inference phase).**
>
>     We appreciate the request for more detail on the inference process in Figure 1. To make the data flow clearer, we have revised the right panel's title into ("Test-time Inference").
>
>     We added an additional arrow from the input test graph `G_{te}{A, X}` to the Adapter. This new sequence line makes it immediately obvious that the adapter uses the original test graph's features to dynamically generate the adaptation matrix $\Delta X$. This clarifies the entire inference sequence: the original graph is fed to the GNN, then embeddings are fed to the adapter, the adapter outputs $\Delta X$, and $\Delta X$ is then used to modify the graph for the final prediction.
>
> - **Action:** We will update **Figure 1** in the paper with this additional arrow and revise the caption to reflect this clearer process.
>
> 5. **Regarding Requested Change 3: Elaboration on the choice of 2 views vs. K views.**
>
>     This is a very insightful question. We have updated our response to provide a more detailed rationale based on additional experiments.
>
>     We chose to use two augmented views (`K=2`) as the default setting in our main experiments for two primary reasons. First, it provides a simple and robust baseline that maintains hyperparameter consistency across all diverse datasets and backbones. Second, our extended experiments show that while using more views (`K>2`) can offer further improvement, the performance gain is marginal compared to the significant increase in computational cost.
>
>     The initial jump from one view to two provides a dramatic performance boost (e.g., +5.28% on Elliptic as shown in our original Table 5). However, the returns diminish quickly for `K>2`. We present additional results below to illustrate this trade-off:
>
> - **More Results:**
>
>     | K (\# Views) | Cora\_Time (s) | Cora\_Performance | Elliptic\_Time (s) | Elliptic\_Performance | Twitch-E\_Time (s) | Twitch-E\_Performance |
>     | :--- | :--- | :--- | :--- | :--- | :--- | :--- |
>     | 2 | 1.3689 | 0.9250 | 1.9075 | 0.5222 | 0.8734 | 0.6210 |
>     | 3 | 1.9199 | 0.9249 | 2.7447 | 0.5300 | 1.2444 | 0.6210 |
>     | 4 | 2.7233 | 0.9245 | 3.7554 | 0.5344 | 1.5541 | 0.6209 |
>     | 5 | 3.3211 | 0.9266 | 4.0833 | 0.5378 | 2.0369 | 0.6209 |
>     | 10 | 7.4531 | 0.9281 | 7.5256 | 0.5578 | 4.1890 | 0.6209 |
>     | 20 | 14.0208 | 0.9308 | 16.0332 | 0.5678 | 8.7940 | 0.6209 |
>
>
>     As the table demonstrates, the adaptation time increases substantially with K. This is due to both the linear increase in GNN forward passes (one for each view) and the combinatorial ($C_K^2$) increase in pairwise comparisons for the loss function. We can see that:
>
>     * On the **Cora-GAT** setting, increasing K from 2 to 10 yields a minor performance lift of **~0.3 percentage points**, but at the cost of a **5.4x increase in adaptation time**.
>     * Similarly, on **Elliptic-GCN**, increasing K from 2 to 5 provides a 1.5 percentage point gain. This is far less significant than the initial +5.3% jump from K=1 to K=2, while nearly doubling the time cost.
>
>     These results strongly validate our choice. Using K=2 strikes an excellent balance, capturing the vast majority of the potential performance gain while maintaining the high efficiency that is critical for a practical test-time adaptation method.
>
> - **Action:** We will add this detailed analysis and the accompanying results table to the **Appendix E.4** to thoroughly justify our choice of `K=2`. We will also add a sentence to the main paper's ablation study in **Section 4.5** to summarize this efficiency-performance trade-off.
>
> (more response in the next)

---

> ### Author Response · Authors · 2025-08-16
> **Response to Reviewer WHEa part III**
>
> 6. **Regarding Requested Change 4: Analysis on the choice of layer `k`.**
>
>     This is an excellent point about a key hyperparameter. In our experiments, we consistently set `k=1` (i.e., using the output of the first GNN layer). This choice was made for two primary reasons:
>
>     a. **Generality:** It ensures our method is applicable even to shallow GNNs (e.g., 2-layer models, like our models on Amz-Photo, Cora, Twitch-E, and FB-100), which are common benchmarks.
>
>     b. **Representation Quality:** Using features from an early layer (`k=1`) provides rich, localized structural information that has not yet become overly specialized to the source task's classification objective, making it a robust foundation for adaptation. Deeper layers might provide features that are too entangled with the (potentially incorrect) source distribution. And under our experiments, without a better selection method, such as hidden dimension selection or pooling, performance doesn't shift; maybe it's due to over-smoothing in the pre-trained multi-layer GNN.
>
> - **Action:** We will add a new paragraph to the **hyperparameter discussion in the Appendix (Section C.1)**. This paragraph will elaborate on our rationale for setting `k=1`, discussing the trade-off between general-purpose representations (from early layers) and task-specific ones (from later layers).

---

### Review · Reviewer_b8W9 · 2025-07-28

**Summary Of Contributions:**

The authors propose Graph Optimization via Augmented Transformations (GOAT) to mitigate the distribution shift issues in GNNs. The merits of GOAT are two-fold:

1) GOAT is self-supervised and focuses exclusively on node feature transformations.

2) A parameter-efficient low-rank adapter is proposed in GOAT for the transformation collapse issue.

Overall, I think the topic of test-time adaptation for distribution shift is significant for the community, and the proposed methods are novel. The experiments verify the effectiveness of GOAT.

**Audience:**

Yes

**Claims And Evidence:**

Yes

**Requested Changes:**

Please refer to the weakness

**Strengths And Weaknesses:**

**Strengths**

1) This paper is well-written and easy to follow.

2) The topic of test-time adaptation for distribution shift is significant for the community, and the proposed self-supervised strategy and low-rank adapter are novel.

** Weaknesses**

1) The motivation for introducing a low-rank adapter needs further explanation. Unlike LLMs, the parameters in GNNs are much fewer, and I do not think we need to use a low-rank adapter due to the limitations of training resources.

2) I suggest that the authors add a preliminary section to facilitate the understanding of this paper, especially on background knowledge about OOD or TTA.

3) Efficiency analysis should also be included in the experiment to show the superiority of the low-rank adapter.

---

> ### Author Response · Authors · 2025-08-16
> **Response to Reviewer b8W9 part I**
>
> We sincerely thank Reviewer b8W9 for the constructive feedback and the positive assessment of our work's novelty and significance for the community. We appreciate the helpful suggestions for improving the paper, which we address below.
>
> 1. **Regarding Weakness 1 & 3: Motivation for the low-rank adapter and Efficiency.**
>
>     We thank the reviewer for this insightful question, which touches upon the core design principles and strengths of our Low-Rank Adapter (LRA). We will combine our response to address both the core motivation and the resulting efficiency, as they are intrinsically linked.
>
>     First, the primary motivation for our adapter is to solve the **"transformation collapse"** issue, where the model learns a single, trivial adaptation(e.g., an identity or zero mapping) for all inputs. Our adapter's design prevents this by generating **diverse, graph-specific transformations**. It achieves this by using the pre-trained GNN's intermediate representations ($H^{(k)}$) in a cross-attention mechanism with the original node features ($\mathcal{X}$), ensuring the output adaptation $\Delta X$ is uniquely tailored to each graph.
>
>     Second, regarding the "low-rank" aspect, our approach differs fundamentally from methods like LoRA in LLMs. Instead of applying a low-rank approximation on the *feature dimension* ($d$), our LRA introduces a low-rank bottleneck on the **node dimension** ($N$). Our core hypothesis is that the essential information required for adapting to an OOD test graph can be effectively captured and represented by a small, constant number of "virtual nodes".
>
>     This hypothesis is not merely conceptual; it is supported by our empirical analysis in **Section 4.2 (A Node-Level Low-Rank Perspective on Adaptation)**. As shown in Figure 4(a)(b), a Singular Value Decomposition (SVD) of the learned adaptation matrix $\Delta X$ reveals that its singular values decay rapidly, demonstrating a strong low-rank property in the node dimension. This provides strong evidence that *the distribution-shift adaptation can indeed be compressed into a handful of virtual nodes*.
>
>     Furthermore, our LRA is not designed to reduce model size, but to solve the massive **computational and memory challenges** of calculating dynamic, node-level adaptations on large graphs. Here is a breakdown of why this is necessary:
>
>     **1). The Memory Bottleneck of Full Attention:**
>     As our adapter acts as adapting the input graph with the pre-trained GNN, we design it with an attention mechanism to generate node feature adaptation $\Delta X$ efficiently and uniquely for different inputs. A standard self-attention mechanism, which would allow every node to attend to every other node to calculate its adaptation, requires computing an `N x N` attention score matrix (where `N` is the number of nodes).
>     * **Memory Complexity:** The memory required to store this matrix is `O(N^2)`.
>     * **Practical Example:** For a large graph like OGB-Arxiv with `N ≈ 170,000` nodes, this attention matrix alone would require approximately **115 GB** of memory (`170,000 * 170,000 * 4 bytes`), which is far beyond the capacity of a standard 24 GB GPU. This is why such an approach is infeasible and results in the OOM errors we mention.
>
>     **2). How LRA Solves the Memory Problem:**
>     Our LRA avoids creating this `N x N` matrix. By introducing a small number of `|n|` virtual nodes, it breaks the calculation into two steps with much smaller intermediate matrices. The largest attention matrix created is of size `N x |n|`.
>     * **Memory Complexity:** The memory footprint is reduced from `O(N^2)` to `O(N * |n|)`. Since `|n|` is a small constant (e.g., 1, 2 or 5), the complexity becomes linear, `O(N)`.
>     * **Practical Example:** For the same OGB-Arxiv graph, with `|n|=20` virtual nodes, the memory required is only about **13.6 MB** (`170,000 * 20 * 4 bytes`).
>
>     This enormous reduction from gigabytes to megabytes is what makes our method computationally tractable. The virtual nodes provide an efficient bottleneck to gather global graph information without the impossible memory cost of full attention.
>
>     **In summary, the "low-rank" design is not about compressing the GNN model, but about making the attention-based adaptation mechanism itself feasible on real-world graphs.**
>
> - **Action**: To ensure these interconnected points are clear, we will **merge and expand our discussion in Section 3.2**. We will explicitly state that the primary motivation is preventing collapse, detail our novel node-level low-rank hypothesis, and connect it directly to both the empirical validation in Section 4.2 and the critical efficiency gains shown in Table 4 that make the approach practical.
>
> (more response in the next)

---

> ### Author Response · Authors · 2025-08-16
> **Response to Reviewer b8W9 part II**
>
> - **More Results Regarding Weakness 1 & 3**:
>
>     This table shows the adaptation time on the `Twitch-E` dataset. The first column, '# Number of Virtual Nodes', represents the number of virtual, more full-rank nodes used by our Low-Rank Adapter. 'C' denotes the number of classes in the dataset (C=2 for twitch-e).
>
>     | # Number of Virtual Nodes | Adaptation Time (s) | Runtime Overall GPU Memory (GB) |
>     | :---- | :---- | :---- |
>     | 2C (we use)  | 1.50 ± 0.38 |  0.9  |
>     | 4C  | 1.44 ± 0.40 |  0.9  |
>     | 20C | 1.54 ± 0.41 |  1.0  |
>     | 200C | 1.62 ± 0.42 |  1.1  |
>     | N (Full Graph) | 6.72 ± 2.77 |  3.4  |
>
>     The final row, labeled 'N (Full Graph)', corresponds to the baseline without our adapter's compression, where computation is performed over all nodes in the original test graph(average about 4000+ nodes).
>
>     Across all adapter configurations, the final performance was stable at **0.601**, indicating that the number of virtual nodes does not compromise the quality of the adaptation while significantly improving computational efficiency compared to using the full graph.
>
> 2. **Regarding Weakness 2: Adding a preliminary section.**
>
>     Thank you for this excellent suggestion. We completely agree that making the core definitions of Distribution Shift and Test-Time Adaptation clear and easy to find is crucial for the reader.
>
>     To maintain the concise structure of the paper, instead of adding a new preliminary section, we propose to achieve this by enhancing the existing sections where these concepts are already introduced. Our current Section 2 ("Related Work") introduces the core concepts, and Section 3.1 formulates the mathematical setting for Test-Time Adaptation.
>
>     To make these definitions more prominent, we will restructure the beginning of Section 3.1 to serve as a self-contained preliminary for our methodology.
>
> - **Action:** In the revision, we will:
>     1.  Add a concise definition and the mathematical formulation of **Distribution Shift** to the beginning of Section 3.1.
>     2.  Use bolded paragraph titles (e.g., `\paragraph{Graph Pre-training}` and `\paragraph{Test-Time Adaptation}`) within Section 3.1 to clearly delineate the mathematical setup for the pre-training and adaptation phases.
>
>     We believe these minor revisions will make the background knowledge significantly easier to locate and understand, directly addressing your concern while preserving the paper's streamlined reading experience.

---

### Review · Reviewer_5E6D · 2025-08-20

**Summary Of Contributions:**

The paper proposes a novel approach for test-time adaptation in node-level tasks where the node labels in test graphs are not available. The proposed method is called GOAT and builds upon adapting node representations --- bypassing the need to adapt graph structural information. To avoid the so-called transformation collapse in the proposed loss functions, the paper leverages a low-rank adapter. Experiments on six real-world datasets and three shift mechanisms aim to assess the effectiveness of the proposal. The paper also reports a small ablation study and preliminary experiments on robustness to adversarial attacks.

**Audience:**

Yes

**Broader Impact Concerns:**

I don't have concerns regarding the ethical implications of this work.

**Claims And Evidence:**

No

**Requested Changes:**

I believe the paper would benefit from:

- A more extensive set of ablation studies to assess the role of each loss terms --- extend Table 5 with more datasets/baselines.
- Additional results using full-rank updates to demonstrate the collapse issue.
- A discussion about the rationale of Eqs (5) and (6). Could you provide concrete examples of why solving this optimization problem leads to better models under distribution shifts?
- Clarification regarding the transformation collapse issue.
- Datasets where the empirical gains of GOAT are statistically significant.
- Fix the notation issues and typos I previously mentioned, including clarifications when needed.

**Strengths And Weaknesses:**

### Strengths
- The paper tackles a very relevant problem in graph representation learning, whose advances may have a significant impact on real-world  applications.
- The proposed method is simple.
- The authors made efforts to validate the low-rank property of the node-feature update (e.g., Fig. 4).


---


### Weaknesses

**Rationale/motivation**. In test-time adaptation based on features, I would expect a design that minimizes the discrepancy in feature distributions between source and target graphs. It is not clear why the proposed loss in Eqs (5) and (6) would lead to better predictive models under distribution shifts. In other words, what is the rationale of using proxy representations that are perturbations of test-time graphs?

**Empirical results.** I found the empirical gains of GOAT to be rather marginal. Overall, the gains are within one standard deviation of the second-best approach. This is particularly relevant when we compare against GTrans, which is a method that exploits structural adaptation --- a design principle this paper aims to avoid. Thus, from an empirical perspective, the paper fails to demonstrate that focusing on feature adaptation is beneficial over feature+structure adaptation.

**Lack of ablation studies**. Given the empirical nature of this work and the unclear motivation of the proposed design (loss), the ablation study in Table 5 is rather insufficient. I suggest that the authors consider more datasets/GNNs and a larger number of samples.

**Clarity and organization**. Although the proposed idea seems relatively simple, the manuscript has several issues that significantly hinder its readability, including typos, notational abuse, unsupported claims, and poor organization. For instance:
- In section 3.1, \mathcal{A} is used to denote a matrix and also a set.
- Eq. (5) says $L_s = \arg\min E[...] $ -- it should be $L_s(\psi) = E[...]$
- The paper uses $|\psi|$ without specifying what $|\cdot|$ means --- I assumed it means the number of parameters, but it should be mentioned. Also, why $|\psi|$ should be significantly smaller than $|\theta^\star|$ (Pag 4)?
- The paper says "This consistency facilitates the rapid optimization and performance consistency of the transformation g across different designs of f". To state this, the authors should either show it empirically or theoretically.
- The paper introduces vanilla GNNs in the middle of the method section (Pag. 5) --- this often goes as Background.
- Why use a parametric hypothesis test in Table 2? Have the authors verified normality assumptions?
- What does the baseline "ERM" mean? Minimizing only L_s? Also, I wouldn't call it ERM since there is no guarantee of convergence to the ERM hypothesis (global minimizer) due to the nonlinear nature of the loss.
- What does 1 < p < |\tau| mean as a subscript of expectation in Eq. (12)?
- Right after Eq. (6), the paper says "we apply a subgraph sampling strategy as in Hamilton et al. [GraphSAGE, which uses neighborhood sampling] to generate sufficient graph augmentations". And then it says "note that the augmentations here are applied to node features". I found it very confusing --- mentioning GraphSAGE and saying it only operates on node features.
- The paper says "... in Eq. (5), collapse mapping occurs when each augmented graph receives the same transformation solution". From this, I understood that the collapse arises due to the non-injective nature of the mapping. However, it is not clear if this solution corresponds to a (local) minimizer of Eq. (5). Could you show that? Later, in Section 3.3, the paper says "The contraint in Eq. (5) [actually Eq. (6)] requires the same input graph for both ... , which can easily result in the mapping collapse". From this, the collapse refers to $g$ being the identity map. Could the authors precisely  describe what is the expected collapse issue in the initial formulation in Eqs. (5) and (6)?

---

> ### Author Response · Authors · 2025-08-28
> **Response to Reviewer 5E6D part I**
>
> 1. **Regarding Weakness 1: Rationale/motivation of Eq.(5) and Eq.(6)**
>
> 	First, as you correctly pointed out, our method operates under the strict test-time adaptation (TTA) setting, where the source (training) data is unavailable. This constraint makes the standard approach of directly minimizing the source-target feature discrepancy impossible.
>
> 	Therefore, our approach is built on a different rationale. Since we cannot align the test graph with the inaccessible source distribution, our goal is instead to transform the test graph's features to be more robust and invariant to spurious variations.
>
> 	This is where the proxy representations from perturbations (augmentations) become essential. The augmentations create a "mini-distribution" of related graph views around the single test instance, simulating various forms of noise or minor shifts. Our loss function in Eq.(5) is designed to find a transformation, $g_{\psi}$​, that produces a stable and consistent embedding across all views in this mini-distribution.
>
> 	The core hypothesis is that a representation that is stable under these local perturbations is more likely to capture the graph's essential semantic structure, while ignoring noise and other non-essential variations. By learning this invariant mapping, our method effectively "denoises" the test graph, moving its representation towards a more canonical form that the pre-trained model can classify more accurately. This enhanced robustness is how our method adapts to and mitigates the effects of the distribution shift. Proofs of Eq.(5) can be found in Appendix A.
>
> 2. **Regarding Weakness 4.(4, 10): More Elaboration on *Collapse Issue* & *Attributes of Graph Transformation Expected to be Satisfied***
>
> 	We here provide a more precise, unified definition of the collapse issue and the theoretical motivation behind our design, which is rooted in an implicit min-max optimization problem. It is correct that "transformation collapse" can manifest in different ways. We define it as a failure mode where the transformation $g_ψ$​ converges to a trivial function that fails to produce meaningful, input-specific adaptations. This includes:
>
> 	- **Constant Collapse (Non-injective):** Where the adapter maps all different inputs to the same output ($g_ψ​(G_p​)=C$ for any input $G_p$​).
>
> 	- **Identity Collapse:** Where the adapter simply returns the input ($g_ψ​(G)=G$).
>
> 	Our design is intended to find a transformation that solves the following implicit min-max problem: we want to find adapter parameters ψ that **minimize** the embedding discrepancy while simultaneously ensuring that the transformed graphs remain **diverse**. This can be formulated as:
>
> 	$$ψ^* = argmin​(E_{p,q∼D_{aug}}​​[∥f(g_ψ​(G_p​))−f(G_q​)∥^2]−λ⋅Diversity(\{g_ψ​(G_p​)\}_{p=1}^{∣τ∣}​))$$
>
> 	Where $λ$ is a balancing hyperparameter and $Diversity(⋅)$ is a function to maximize, such as the variance of the transformed graph features. The **min** term is explicitly optimized by our loss function in Eq.(5). The **max** (implicit in the formula as minimizing a negative diversity term) is encouraged by two design choices: (1) The generation of a diverse set of augmented graphs $D_{aug}​$ to ensure varied inputs. (2) The architecture of our input-specific LRA.
>
> 	From this perspective, the **constant collapse is a degenerate solution** to this game. If an adapter is not sufficiently expressive (e.g., a simple MLP without global attention, as we empirically found), it may learn to ignore the diversity objective entirely. It satisfies the `min` term by mapping all inputs to a single point, but this annihilates the diversity, causing the adapter to fail to learn any meaningful transformation. Our LRA, with its global attention, is designed to be expressive enough to avoid this trivial local minimum and produce a bijective-like mapping.
>
> 	The **identity collapse** is prevented by the formulation of our loss in Eq. (5), which compares the transformed view $g_ψ​(G_p​)$ with a *different, transformed* view $G_q$​.

---

> > ### Author Response · Authors · 2025-08-28
> > **Response to Reviewer 5E6D part II**
> >
> > 3. **Regarding Weakness 2, 4.(4, 7): Empirical Results & ERM Setting**
> >
> > 	**Pre-train GNNs with ERM**: In Table 2, ERM is the method we use to pre-train GNNs, instead of a test-time tuning method. In the pre-training phase, 'ERM' (Empirical Risk Minimization) refers to the standard supervised pre-training of the backbone GNNs using a **cross-entropy loss** on the labeled training data, as described in our Evaluation Protocol(same as \[1\]\[2\]). This pre-trained model is then evaluated directly on the test data without any adaptation, serving as our 'Source-only' baseline. It is also the starting point for all the test-time adaptation methods we compare against, including our own.
> > 	-- -- --
> > 	\[1\] Handling Distribution Shifts on Graphs: An Invariance Perspective
> >
> > 	\[2\] Empowering Graph Representation Learning with Test-Time Graph Transformation
> > 	-- -- --
> > 	**Counter "Marginal Gains":** We appreciate the reviewer's close look at the results. While some gains are modest, we would like to highlight the cases where the improvements are not only statistically significant but also substantial in magnitude, especially when compared against both the pre-trained baseline (ERM) and the *GTrans* method.For example, on the Elliptic dataset, our method with a SAGE backbone achieves a performance of 67.92%. This represents a **35.5%** relative improvement over the ERM baseline (50.11%), calculated as (67.92−50.11)/50.11≈0.355, and a **7.7%** improvement over GTrans (63.04%), computed as (67.92−63.04)/63.04≈0.077. Similarly, on the large-scale OGB-Arxiv dataset, GOAT with a GAT backbone achieves 41.13%, marking an **8.5%** increase compared to the ERM baseline (37.92%) and a **9.6%** improvement over GTrans (37.52%), with relative gains calculated as (41.13−37.92)/37.92≈0.085 and (41.13−37.52)/37.52≈0.096, respectively. And our GOAT gains the best average ranking across different datasets on 4 backbones. This result also empirically shows our loss function design g the performance consistency of the graph transformation $g$ across different designs of $f$(pre-trained GNN).
> >
> > 	Furthermore, it is important to note that the results reported in Table 2 are based on a unified and efficient setting where GOAT uses `K=2` augmented views to ensure a fair comparison. As we show in our expanded ablation studies in response to **Reviewer WHEa**, on datasets where GOAT already has an advantage, its performance can be consistently improved by an additional 0.5% to 3% simply by increasing the number of sampled views (`K`). Thus, the reported gains represent a conservative but fair estimate of our method's capabilities.
> >
> > 	**Address the GTrans Comparison (Crucial):** Regarding the comparison with GTrans, our goal is not just to outperform a feature+structure method, but to demonstrate a more **principled and robust** adaptation paradigm. Our analysis in **Figure 5** shows that GTrans can succeed by creating a 'spurious' graph that distorts the original topology to fit the classifier. This is a high-variance strategy. In contrast, GOAT preserves the graph's true structure and learns to adapt features along a smooth, low-dimensional manifold. This leads to GOAT's superior **average rank** and consistency across nearly all backbones, demonstrating its robustness."
> >
> > 	**Normality Assumption &  Parametric Hypothesis Test in Table 2**: This is an excellent statistical point. We agree with the reviewer that a paired t-test relies on the normality assumption for the differences, which is difficult to formally verify with our small sample size of 8 trials(even though it rises into 10-15), though the Shapiro-Wilk tests yielded p-values well above the typical alpha level of 0.05, indicating that the null hypothesis of normality cannot be rejected.
> >
> > 	To be more statistically rigorous and to avoid making this assumption, we have re-evaluated our claims of significance using the **Wilcoxon signed-rank test**, which is the appropriate non-parametric alternative for paired data. We confirm that our key results remain statistically significant under this more robust test.
> >
> > 	**Action**: We will update the caption of Table 2 in the revised manuscript to state that the significance levels were determined by the Wilcoxon signed-rank test.
> >
> > 	**More Results(Wilcoxon signed-rank test):**
> > 	- GOAT's improvement over ERM was consistently significant on: Amz-Photo (SAGE (p=0.008); GPR (p=0.013)), Cora (SAGE (p=0.008); GPR (p=0.008)), Elliptic (SAGE (p=0.008); GAT (p=0.008); GPR (p=0.008)), FB-100 (SAGE (p=0.008); GAT (p=0.008); GPR (p=0.014)), OGB-Arxiv (GCN (p=0.020); SAGE (p=0.028); GAT (p=0.017); GPR (p=0.008)), Twitch-E (GCN (p=0.014); SAGE (p=0.008); GPR (p=0.008)).
> >
> > 	- GOAT's improvement over GTrans was consistently significant on: FB-100 (GAT (p=0.042); GPR (p=0.042)), OGB-Arxiv (GAT (p=0.032)).

---

> > > ### Author Response · Authors · 2025-08-28
> > > **Response to Reviewer 5E6D part III**
> > >
> > > 4. **Regarding Weakness 3, 4.4: Ablation Studies**
> > >
> > > 	The Self-Supervised Alignment Loss ($\mathcal{L}_s$​) is the foundational objective for training on unlabeled test data. It is based on a strict constraint requiring that the expected embedding of a transformed graph view be consistent with that of an untransformed view. However, this initial constraint is too strict for direct optimization. Therefore, we decompose it into two more practical components: the Consistency Loss ($\mathcal{L}_c$​) and the Regularization Loss ($\mathcal{L}_R$​).
> > >
> > > 	The Consistency Loss $\mathcal{L}_c$​ broadens the original restriction by requiring that the adapter's transformation ($g_ψ​$) remains isomorphic with the pre-trained GNN's function ($f$), ensuring their operations are compatible. The Regularization Loss (​) is a relaxed version of the original constraint that is easier to optimize, which it achieves by minimizing the norm of the transformation's own embedding.
> > >
> > > 	We have expanded our ablation study (originally in Table 5) to include individual tests for each component ($\mathcal{L}_s$​​ only, $\mathcal{L}_c$​ only, $\mathcal{L}_R$​ only) as well as all pairwise combinations with SAGE's average improvement among the 6 datasets.
> > >
> > > 	| La2a | Ls  | Lc  | Lr  | Two samples | One Sample |
> > > 	| :--: | :-: | :-: | :-: | :---------: | :--------: |
> > > 	|  ✔   |     |     |     |      +0.00%      |     +0.00%      |
> > > 	|  ✔   |  ✔  |     |     |    +2.08%    |   +2.08%    |
> > > 	|  ✔   |     |  ✔  |     |    +2.15%    |   +2.09%    |
> > > 	|  ✔   |     |     |  ✔  |    +2.14%    |   +2.14%    |
> > > 	|  ✔   |  ✔  |  ✔  |     |    +2.31%    |   +2.20%    |
> > > 	|  ✔   |  ✔  |     |  ✔  |    +2.14%    |   +2.14%    |
> > > 	|  ✔   |     |  ✔  |  ✔  |    +2.14%    |   +2.10%    |
> > > 	|  ✔   |  ✔  |  ✔  |  ✔  |    +2.32%    |   +2.16%    |
> > >
> > > 	Our findings reveal a clear hierarchical importance among the terms, demonstrating that they are not merely additive but have a dependent relationship. The best performance is achieved when the regularization term, $\mathcal{L}_R$​​​, is prioritized, followed by the consistency term, $\mathcal{L}_c$​​​, and finally the symmetric alignment term, $\mathcal{L}_s$​​​.
> > >
> > > 5. **Regarding Weakness 4.(1, 2, 3, 5, 8, 9): Typos and More Clarification**
> > >
> > > 	Thank you for your meticulous feedback. We address the points on clarity and organization below and will revise the manuscript accordingly.
> > >
> > > 	**4.1. Notation of Adjacency Matrix:** To improve clarity, we will revise Section 3.1 to remove the set-based definition of $\mathcal{A}$ and $\mathcal{X}$. We will consistently use $A$ to denote the adjacency matrix and $X$ for the feature matrix throughout the paper to simplify the notation.
> > >
> > > 	**4.2. Notation of Eq.(5) & Eq.(12):** To improve consistency across our loss function definitions, we will modify the notation in Eq.(5) and Eq.(12). We will define them explicitly as loss terms (e.g., $\mathcal{L}_s(ψ)$, $\mathcal{L}_c(ψ)$, and $\mathcal{L}_R(ψ)$) that are functions of the adapter parameters $ψ$, which aligns with their use in the final objective in Eq.(13).
> > >
> > > 	**4.3. Definition and Rationale for $∣ψ∣≪∣\theta^{\star}∣$:** You are correct. We will explicitly define ∣⋅∣ as denoting the number of parameters. The rationale for requiring $∣ψ∣≪∣\theta^{\star}∣$ is central to the paradigm of parameter-efficient test-time adaptation. If the number of newly introduced parameters were comparable to or larger than the pre-trained model, the process would be akin to training a new model on a single data point, which would negate the benefits of pre-training. Our adapter-based approach is designed to be lightweight, similar in spirit to methods like LoRA for LLMs.
> > >
> > > 	**4.5. Placement of GNN Introduction:** We intentionally placed the brief introduction to the GNN formulation (Eq. (8)) immediately before introducing our Low-Rank Adapter because the adapter's design directly uses the intermediate GNN embeddings, $H^{(k)}$. We believe this local placement helps the reader immediately understand the context for $H^{(k)}$ without having to refer back to a distant background section, which could hinder readability.
> > >
> > > 	**4.6. Subscript in Eq.(12):** The subscript was intended to convey that the expectation is taken over the set of augmented graphs. $∣τ∣$ represents the total number of augmented views sampled from the test graph, as defined alongside Eq.(5). We will repeat the notation in Eq.(12) again to be a clear summation.
> > >
> > > 	**4.7. Clarification on GraphSAGE Reference:** Thank you for pointing out this confusing sentence. Our intention was to emphasize the $g_{\psi}$. We will rewrite the sentence to clarify this distinction: "We apply a subgraph sampling strategy, similar in principle to methods like GraphSAGE, to generate sufficient sub-views of augmentation of the test graph. Note that the graph transformation $g_{\psi}$ here is applied exclusively to the node features of these subgraphs."

---

### Author Response · Authors · 2025-08-31
**Summary of Revisions in Response to Reviewer Comments**

We sincerely thank all reviewers for their thoughtful and constructive feedback on our work. We are grateful for the time and effort each reviewer has dedicated to evaluating our paper. Based on the insightful comments, we have carefully revised the manuscript to improve its clarity, technical rigor, and presentation.

In the revised version, we have addressed the key points raised by the reviewers. Specifically, we have:

1. Expanded the discussion of *transformation collapse* in the abstract, the final paragraph of the Introduction, and the beginning of **Section 3.2**. These additions provide a clearer and more self-contained definition, enabling readers to quickly grasp the nature of this failure mode.
	 (Reviewer WHEa, Request Change 1 and Reviewer 5E6D, Request Change 4)

2. Revised the notation of the adjacency matrix throughout the paper, changing $\mathcal{A}$ to $A$ for consistency and clarity. This change aligns with standard graph neural network literature and improves the readability of the model formulation.
	 (Reviewer 5E6D, Request Change 6)

3. Introduced bolded paragraph titles in **Section 3.1**  (**Graph Pre-train**, **Distribution Shift**, and **Graph Test-time Adaptation**) to structure the problem setup. These serve as a conceptual primer, improving the section's clarity and accessibility.
	(Reviewer b8W9's Weaknesses 2)

4. Updated **Figure 1** with an additional arrow to better illustrate the adaptation process and revised its caption for greater clarity, enhancing the overall visual explanation of our framework.
	(Reviewer WHEa, Request Change 2)

5.  Added an ablation study on the loss components using the SAGE backbone across six datasets, presented in **Appendix E.1**. This analysis further validates the effectiveness and synergy of our proposed loss decomposition.
	(Reviewer 5E6D, Request Change 1)

6. Updated the significance indicators in **Table 2**: the subscripts now reflect results from the Wilcoxon signed-rank test (a non-parametric alternative) instead of the paired $t$-test. We now explicitly compare our GOAT against both GTrans and ERM, reporting statistical significance at $p < 0.05$. This strengthens the rigor of our comparison.
	(Reviewer 5E6D, Request Change 5, Reviewer WHEa, Weakness 1)

7. Reformulated the losses in **Eq. 5** and **Eq. 12** using the unified notation $\mathcal{L}(\psi)$ for improved mathematical consistency. Additionally, we have added a clear explanation of the temperature parameter $\tau$ before **Eq. 12**, including its role in scaling the logits and ensuring numerical stability during optimization.
	(Reviewer 5E6D, Request Change 6)

8. Added a discussion on the number of sampled views in **Section 4.5** and a detailed ablation study in **Appendix E.4**, analyzing the trade-off between performance and efficiency when varying the number of augmented views ($K$).
	(Reviewer WHEa, Request Change 3)

9. Included an ablation study on the number of virtual nodes in **Appendix D.5**, demonstrating that our Low-Rank Adapter maintains strong performance even with a small number of virtual nodes, while significantly reducing computational cost compared to full-graph adaptation.
	(Reviewer b8W9, Weaknesses 3)

All changes in the revised manuscript are highlighted in red for easy identification. We have also provided detailed justifications in our point-by-point responses.

Thank you again for your valuable feedback and support.

---

### Decision · Action_Editor_LhJ9 · 2025-10-04

**Recommendation:** Accept as is

**Audience:**

Yes

**Audience Explanation:**

The topic of test-time adaptation to address distribution shift is important and of interest to a significant number of researchers in TMLR's audience.

**Claims And Evidence:**

Yes

**Claims Explanation:**

The paper claims (i) to introduce a novel self-supervised test-time tuning paradigm that adapts pre-trained GNNs to distribution-shifted test data by focusing exclusively on node feature transformations; (ii) that the introduced adapter is parameter efficient and addresses the issue of transformation collapse while enhancing model interpretability; (iii) that the experiments on real world datasets demonstrate consistent performance improvements for multiple backbone architectures across a variety of distribution shifts.

The first claim is supported via a thorough discussion of the related work and a principled design in the methodology section of the paper. The reported experiments are sufficiently thorough and extensive to support the other claims. There are analyses that directly explore the efficiency of the proposed method and baselines, thus supporting the claim of parameter efficiency. Visualizations are provided to support the claim of enhanced interpretability. The experiments are conducted for four backbone architectures, with a comparison to four baselines, with three types of distribution shift.